# An improved land biosphere module for use in the DCESS Earth System Model (Version 1.1) with application to the last glacial termination

Roland Eichinger[1], Gary Shaffer[2,3], Nelson Albarrán[4], Maisa Rojas[1], and Fabrice Lambert[5]

[1]Department of Geophysics, University of Chile, Blanco Encalada 2002, Santiago, Chile
[2]GAIA-Antarctica, University of Magellanes, Avenida Bulnes 01855, Punta Arenas, Chile
[3]Niels Bohr Institute, University of Copenhagen, Blegdamsvej 17, Copenhagen, Denmark
[4]Department of Physics, University of Santiago de Chile, Avenida Ecuador 3493, Santiago, Chile
[5]Department of Physical Geography, Catholic University of Chile, Vicuña Mackenna 4860, Santiago, Chile

*Correspondence to:* Roland Eichinger (roland@dgf.uchile.cl)

**Abstract.**

Interactions between the land biosphere and the atmosphere play an important role for the Earth's carbon cycle and thus should be considered in studies of global carbon cycling and climate. Simple approaches are a useful first step in this direction but may not be applicable for certain climatic conditions. To improve the ability of the reduced-complexity Danish Center for Earth System Science (DCESS) Earth System Model DCESS to address cold climate conditions, we reformulated the model's land biosphere module by extending it to include three dynamically varying vegetation zones as well as a permafrost component. The vegetation zones are formulated by emulating the behavior of a complex land biosphere model. We show that with the new module, the size and timing of carbon exchanges between atmosphere and land are represented more realistically in cooling and warming experiments. In particular, we use the new module to address carbon cycling and climate change across the last glacial transition. Within the constraints provided by various proxy data records, we tune the DCESS model to a Last Glacial Maximum state and then conduct transient sensitivity experiments across the transition under the application of explicit transition functions for high latitude ocean exchange, atmospheric dust, and the land ice sheet extent. We compare simulated time evolutions of global mean temperature, $pCO_2$, atmospheric and oceanic carbon isotopes as well as ocean dissolved oxygen concentrations with proxy data records. In this way we estimate the importance of different processes across the transition with emphasis on the role of land biosphere variations and find that carbon outgassing from permafrost and uptake of carbon by the land biosphere broadly compensate each other during the temperature rise of the early last deglaciation.

## 1  Introduction

On centennial to millennial time scales, ocean processes may largely determine variations of atmospheric $CO_2$ concentrations (Fischer et al., 2010; Sigman et al., 2010). Such processes include changes in ocean dynamics as well as in biogeochemical properties like variations in the phosphate inventory or iron fertilisation (Martin et al., 1990; Maher et al., 2010). However, also interactions between atmosphere and land can have an important impact on the overall change in the carbon cycle and

thus on the Earth's climate system. Net primary production on land takes up $CO_2$ from the atmosphere at a rate that increases with the $pCO_2$ itself ($CO_2$ fertilisation; Saugier et al., 2001). Remineralisation in the soils increases with increasing temperature (Davidson and Janssens, 2006). Different vegetation zones advance and retreat due to varying climate conditions, thereby changing the terrestrial biomass budget and thus the carbon amount being stored in vegetation (Ciais et al., 2012). Moreover,

changes in permafrost area and, during glacial conditions, changes in areas covered by ice sheets also have the potential to significantly modify atmospheric $pCO_2$ (Schuur et al., 2008). The release of carbon into the atmosphere through the thawing of permafrost in a warming future climate has been assessed in a number of studies (e.g. Schaefer et al., 2011; Schuur et al., 2008; Khvorostyanov et al., 2008) and carbon storage and release in and from permafrost can also help explain glacial-interglacial cycles (Zech, 2012; Ciais et al., 2012; Crichton et al., 2016). A land biosphere module within an Earth System Model should

be able to address these processes.

For this reason, we here extend the Danish Center for Earth System Sciences (DCESS) Earth System Model (Shaffer et al., 2008) by a new terrestrial biosphere scheme. Our new module features the three vegetation zones, tropical forests (TF), grasslands-savanna-deserts (GSD) and extratropical forests (EF), through definition of their characteristic values of biomass

reservoirs and net primary production (NPP). The dynamic accounting of the latitudinal boundaries of the different zones and thereby their area extents is approximated by fitting polynomial functions of global mean temperature ($T_{glob}$) to results of a complex vegetation model study by Gerber et al. (2004). For completeness we also developed a simple approach to vegetation albedo based on the relative sizes of the three vegetation zones. Moreover, we present a component that accounts for carbon being stored in permafrost and below terrestrial ice sheets to allow extensive carbon storage on land during glacial climate con-

ditions and its release across deglaciation events. In DCESS model simulations, these new developments considerably improve the estimates of amount and timing of land-atmosphere carbon exchanges, including the carbon isotopes [13]C and [14]C.

For a first application of this new module, we furthermore developed a set of explicit functions that describe the transitions of high latitude ocean exchange, atmospheric dust and land ice sheet extent across the last 25 kaBP. This allows us to simultaneously simulate time series of global mean temperature, $pCO_2$, atmospheric and oceanic carbon isotopes as well as

ocean dissolved oxygen concentrations across the deglaciation after the Last Glacial Maximum (LGM, ∼21,000 years ago). Hitherto, the DCESS model has been used mainly for future climate projections (see e.g. Shaffer et al., 2009; Shaffer, 2010) and evaluated for pre-industrial (PI) climate conditions (see Shaffer et al., 2008). For the present application, the model is calibrated to glacial conditions by adapting physical and biogeochemical parameters guided by proxy data records. This includes a physically simple method to generate isolated deep water in the high latitude model ocean (as it had been hypothesised by

several studies, e.g. Francois et al., 1997; Sigman and Boyle, 2000; Broecker and Barker, 2007) through the imposition of a depth profile for the vertical exchange intensity. Transient sensitivity simulations across the last 25 kaBP are then performed. These demonstrate the impact and timing of various processes on atmospheric temperatures, $pCO_2$ and the carbon isotopes [13]C and [14]C at the beginning of the last glacial termination ("Mystery Interval" (MI), from 17.5 to 14.5 kaBP; Broecker and Barker, 2007).

## 2  A new land biosphere module for the DCESS model

The DCESS model features components for the atmosphere, ocean, ocean sediment, land biosphere and lithosphere and has been designed for global climate change simulations on time scales from years to millions of years (Shaffer et al., 2008). Its geometry consists of one hemisphere, divided into two 360° wide zones by 52° latitude. The model ocean is divided into a low-mid and a high latitude sector (as in the HILDA (**hi**gh-**l**atitude exchange/interior **d**iffusion **a**dvection) model, developed by Shaffer and Sarmiento, 1995) and features a continuous vertical resolution of 100 m, to a depth of 5500 m. The near surface atmospheric mean temperature is described by a simple, zonal mean, energy balance model in combination with sea ice and snow parameterisations. The atmosphere is assumed to be well mixed for gases and air-sea gas exchange fluxes and transports via weathering, volcanism and interactions with the land biosphere are considered for carbon dioxide ($CO_2$) and methane ($CH_4$) in $^{12,13,14}C$ species, respectively, as well as for nitrous oxide ($N_2O$) and oxygen ($O_2$). Ocean dynamics are characterised by high latitude sinking and low-mid latitude upwelling as well as horizontal and vertical diffusion between the latitude zones and the ocean layers. For the ocean biogeochemical cycling, a number of tracers are considered (namely, phosphate ($PO_4$), dissolved oxygen ($O_2$), dissolved inorganic carbon ($DI^{12,13,14}C$), and alkalinity (ALK)), which are forced by new production, air-sea exchange, remineralisation of organic matter, dissolution of $CaCO_3$, river inputs and evaporation/precipitation (Shaffer, 1996; Shaffer et al., 2008). There is a sediment section for each of the ocean model layers addressing $CaCO_3$ dissolution/burial and organic matter remineralisation/burial.

A land biosphere scheme accounts for the $^{12,13,14}C$ cycling with leaf, wood, litter and soil boxes (Shaffer et al., 2008). NPP on land takes up $CO_2$ from the atmosphere and is distributed between leaves and wood. Leaf loss goes to litter, wood loss is divided between litter and soil and litter loss is divided between the atmosphere (as $CO_2$) and the soil. Soil loss goes to the atmosphere as $CO_2$ and $CH_4$. Losses from all land reservoirs are taken to be proportional to reservoir size and, for litter and soil, to depend upon the mean atmospheric temperature according to $\lambda_Q \equiv Q_{10}^{(T_{glob}-T_{glob,PI})/10}$, where $Q_{10}$ (a biotic activity increase for a 10 degree increase of $T_{glob}$) is chosen to be 2 (Friedlingstein et al., 2006).

When formulated in this simplistic manner, our original biosphere module does not take into consideration changes in vegetated area and thereby overestimates land biosphere biomass for cold conditions (see below). In an attempt to remedy these deficiencies while retaining simplicity on the level of the rest the model, we here present the extension of this scheme to three different vegetation zones. We define a latitudinal distinction of these three vegetation zones and their latitudinal boundaries on a global scale. The zones we consider are tropical forests (TF), grasslands, savanna and deserts (GSD) and extratropical forests (EF) containing temperate and boreal forests. In this section, we first present the characteristics of the chosen vegetation zones and their latitudinally variable borders. Then, the new calculations of the biosphere-atmosphere exchange fluxes of $CO_2$ and $CH_4$ for $^{12}C$ as well as for the rare carbon isotopes $^{13}C$ and $^{14}C$ are described and a simplified formulation of the treatment of permafrost is given. Moreover, in this section, we provide a brief evaluation of the new vegetation module, to show how it represents land-atmosphere carbon fluxes on centennial to millennial time scales.

## 2.1 Description of the vegetation zones

The three vegetation zones (TF, GSD, EF) were defined on the basis of a study by Gerber et al. (2004). In that study, the complex LPJ terrestrial biosphere model (Lund-Potsdam-Jena Dynamic Global Vegetation Model) was applied to distinguish between a number of vegetation zones based on several variables. The latitudinal limits of these vegetation zones are dynamically defined. In general, the extent of certain vegetation zones depends mainly on temperatures and precipitation. However, the limitations of the DCESS model (no explicit computation of precipitation and restriction to two latitudinal sections) require a somewhat more general approach. We therefore determine the division of the three vegetation zones solely by the deviation of the global mean atmosphere temperature from its PI value ($15°C$). For this purpose, we derived two polynomial functions from a study by Gerber et al. (2004). We started from the total tree cover frame of their Fig. 4 by reading off, at $2°C$ intervals from -10 to $10°C$ deviation from pre-industrial global mean temperature, the latitudes in the Northern Hemisphere of 50% tree cover both above and below the subtropical zone of lower tree cover. Each of these two sets of 11 points formed the basis of our curve fitting. We found that 5th order polynomials provided good fits to each of these sets. This emulation of a complex vegetation model thereby implicitly includes the role of precipitation in the temperature-dependence of the vegetation zone boundaries. The two latitudinal limitations of the vegetation zones are described by the two 5th order polynomials

$$L_{TF-GSD} = -1.83 \cdot 10^{-5} \cdot \delta T_{glob}^5 - 0.0005809 \cdot \delta T_{glob}^4 - 0.005168 \cdot \delta T_{glob}^3 \tag{1}$$

$$+ 0.0497 \cdot \delta T_{glob}^2 + 1.092 \cdot \delta T_{glob} + 11.28 \tag{2}$$

and

$$L_{GSD-EF} = 1.152 \cdot 10^{-5} \cdot \delta T_{glob}^5 - 0.0001785 \cdot \delta T_{glob}^4 - 0.004557 \cdot \delta T_{glob}^3 \tag{3}$$

$$+ 0.04156 \cdot \delta T_{glob}^2 + 1.017 \cdot \delta T_{glob} + 37.77, \tag{4}$$

which depend only on the deviation of the global mean atmosphere temperature $\delta T_{glob}$ from the calibrated PI steady-state. $L_{TF-GSD}$ denotes the latitude of the border between the TF and the GSD zones and $L_{GSD-EF}$ the latitude between GSD and EF. These two 5th order polynomials are illustrated in Fig. 1.

5      The EF vegetation zone additionally is limited by either the model snowline or the line of the terrestrial ice sheet extent, depending on which one of the two lines expands the farthest from the pole at the current time step (see Sect. 2.4 for definition of "snowline" and further explanations). The snowline is also included in Fig. 1, the zone poleward of of the snowline is taken to be permafrost area in our simplified approach. Based on these latitudinal limits, the total $CO_2$ and $CH_4$ fluxes between the terrestrial biosphere and the atmosphere are now determined by the sum of the three vegetation zones and thereby depend on

10     the areas and mean temperatures of each zone as well as their values of NPP and stored biomass.

     Table 1 shows the characteristic values of biomass reservoirs and NPP of those vegetation zones at PI climate conditions ($T_{glob} = 15°C$, $p$CO$_2$=280 ppm) (Gower et al., 1999; Saugier et al., 2001; Sterner and Elser, 2002; Zheng et al., 2003; Chapin et al., 2011). The values in table 1 have been constrained such that the sum over the three vegetation zones adds up to global PI values of the original biosphere model (Shaffer et al., 2008).

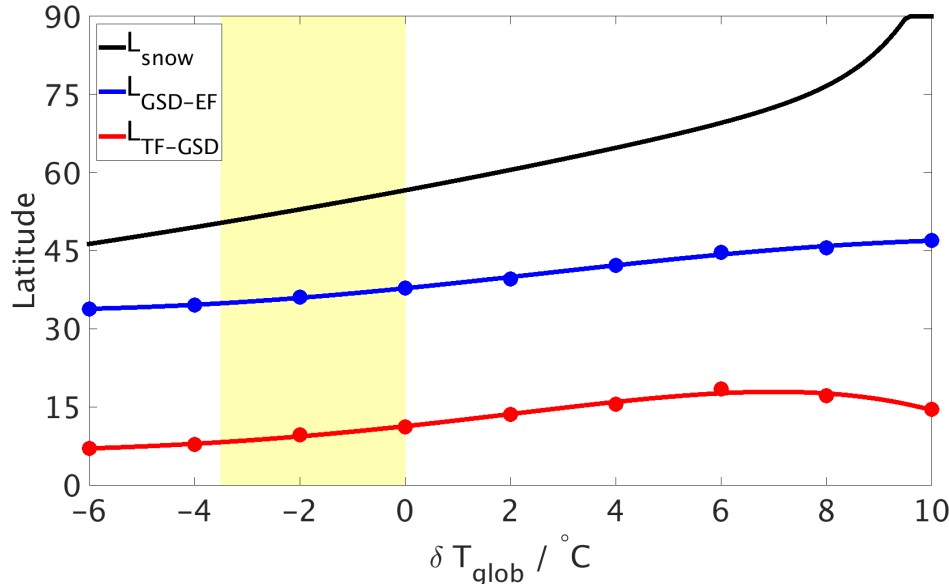

**Figure 1.** Polynomial functions describing the dynamic latitudes of the borders between the three vegetation zones as function of the global mean atmosphere temperature ($\delta T_{glob}$) deviation and the latitude of the "snowline" (black). Red: Border between the TF and the GSD zone ($L_{TF-GSD}$). Blue: Border between the GSD and the EF zone ($L_{GSD-EF}$). The dots mark the points from the curve fitting as discribed in the text. The yellow bar marks the region between LGM and PI climate conditions. (PI: $\delta T_{glob,PI} = 0°$C, $L_{TF-GSD,PI} = 11.28°$, $L_{GSD-EF,PI} = 37.77°$, $L_{snow,PI} = 55°$); LGM: ($\delta T_{glob,LGM} = -3.5°$C; $L_{TF-GSD,LGM} = 7.17°$; $L_{GSD-EF,LGM} = 33.92°$, $L_{snow,LGM} = 51°$)

|  | Tropical forest | Grassland savanna desert | Extratropical forest |
|---|---|---|---|
| Leaves / $GtC$ | 30 | 20 | 50 |
| Wood / $GtC$ | 270 | 180 | 50 |
| Litter / $GtC$ | 16 | 64 | 40 |
| Soil / $GtC$ | 200 | 800 | 500 |
| NPP / $Gt \cdot a^{-1}$ | 25 | 15 | 20 |
| Area / $10^6 km^2$ | 25 | 53 | 27 |

**Table 1.** Pre-industrial distribution of carbon storage among model land carbon pools as well as model net primary production for the three vegetation zones (see Chapin et al., 2011, and citations therein).

## 2.2 Vegetation albedo

For completeness and consistency, we also extended the model albedo calculation to account for the new biosphere scheme with the three vegetation zones. In the DCESS model, albedo, $\alpha$, is taken to be constant and equal to 0.62 for all snow or ice covered areas. For non-snow/ice covered areas, $\alpha$ is expressed as

$$\alpha = a + b \cdot \left\{ 0.5 \cdot (3 \cdot sin\Theta)^2 - 1 \right\} \tag{5}$$

where $\Theta$ is the latitude, $b = 0.175$ and $a = 0.3$ for present day conditions. This functional form and these constant values have been based on present day observations (Hartmann, 1994). The albedo of non-snow/ice covered areas should vary with vegetation type since forested areas have lower albedo than non-forested areas (Bonan, 2008). As seen in Fig. 1, as the Earth cools from present day, both forested model areas (EF and TF zones) contract while the non-forested model area (GSD zone) expands slightly, in part in response to dryer conditions (Gerber et al., 2004). This would lead to higher albedo and a positive feedback on the cooling. For completeness in our new treatment of the role of the land biosphere in climate and to capture such albedo variations within the context of our new land biosphere module, we assume that a in Eq. 5 may be related to vegetation type such that

$$a = 0.3 - \gamma \cdot \left( 1 - \frac{frac(\delta T_{glob})}{frac_0} \right) \tag{6}$$

where the factor 0.3 is the present day value of a, $\gamma$ is a multiplier, the value of which is determined by calibration (see below), $frac$ is the ratio of the area of the GSD zone to the total non-snow/ice covered area (i.e. the sum of the areas of the EF, GSD and TF zones) and $frac_0$ is this ratio for present day. Note that $frac(\delta T_{glob})$ can be taken from Fig. 1 or calculated explicitly using Eqs. 5 and 6 and the snowline/ice sheet dependency on $\delta T_{glob}$ (see Fig. S3 in the Supplement). Fig. 2a shows a plot of $frac(\delta T_{glob})/frac_0$.

The vegetation albedo forcing for the LGM ($\delta T_{glob} = -3.5°C$) relative to present day has been determined in more complex models from which we choose the value of $-0.7\,\text{W/m}^2$ as being representative (Köhler et al., 2010). Together with Eq. 6 and the model latitudinal distribution of solar forcing, we find that this LGM vegetation albedo forcing anomaly is obtained in our model simulation for a $\gamma$ value of 0.02, a value we adopt here. Fig. 2b illustrates new albedo distributions with latitude for the specific cases of $\delta T_{glob} = -4$, 0 and 4°C for which $a = 0.3027$, 0.3 and 0.2976, respectively.

## 2.3 Extension of the carbon flux equations

In the original version of the DCESS terrestrial biosphere module (Shaffer et al., 2008), the global vegetation NPP is determined by

$$NPP = NPP_{PI} \left( 1 + f_{CO2} \cdot ln \left( \frac{pCO_2}{pCO_{2,PI}} \right) \right). \tag{7}$$

Now, we subdivide this equation into three equations

$$NPP_{TF} = NPP_{TF,PI} \cdot A_{TF} \cdot \left( 1 + f_{CO2} \cdot ln \left( \frac{pCO_2}{pCO_{2,PI}} \right) \right), \tag{8}$$

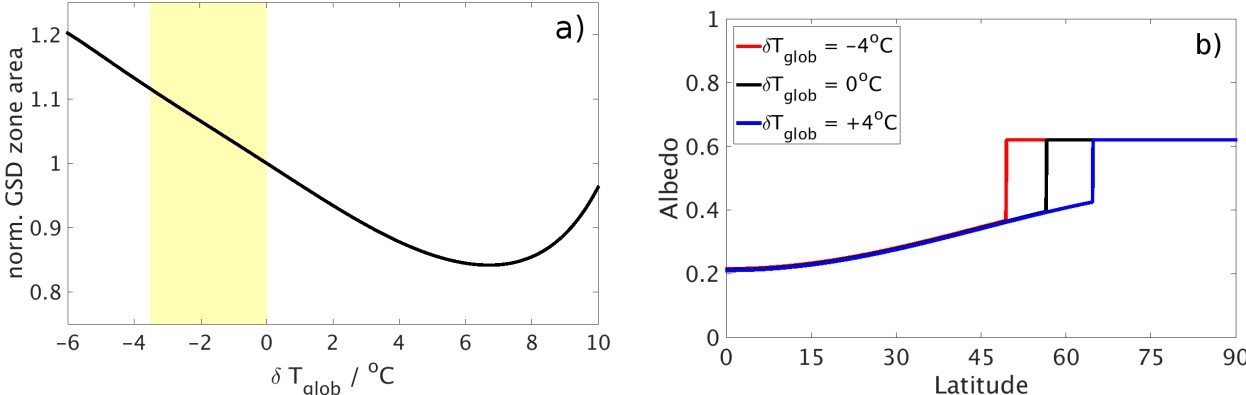

**Figure 2.** a) Normalised GSD zone area fraction as function of global mean temperature deviation from PI climate conditions. The yellow bar marks the region between LGM and PI. b) Latitude dependency of albedo for three different deviations from the global mean temperature ($-4°C$, $0°C$ and $4°C$). Note that poleward of the snow line the albedo is 0.62 (albedo of snow/ice covered area).

$$NPP_{GSD} = NPP_{GSD,PI} \cdot A_{GSD} \cdot \left(1 + f_{CO2} \cdot ln\left(\frac{pCO_2}{pCO_{2,PI}}\right)\right) \tag{9}$$

and

$$NPP_{EF} = NPP_{EF,PI} \cdot A_{EF} \cdot \left(1 + f_{CO2} \cdot ln\left(\frac{pCO_2}{pCO_{2,PI}}\right)\right) \tag{10}$$

5    for the different vegetation zones, respectively. Thus, the global $NPP$ is now determined by the sum of the $NPP$ of the three vegetation zones:

$$NPP = NPP_{TF} + NPP_{GSD} + NPP_{EF} \tag{11}$$

The factors $A_{TF}$, $A_{GSD}$ and $A_{EF}$ are calculated by

$$A_{TF} = \frac{sin(L_{TF-GSD})}{sin(L_{TF-GSD,PI})}, \tag{12}$$

$$A_{GSD} = \frac{sin(L_{GSD-EF} - L_{TF-GSD})}{sin(L_{GSD-EF,PI} - L_{TF-GSD,PI})} \tag{13}$$

and

$$A_{EF} = \frac{sin(L_s) - sin(L_{GSD-EF})}{sin(L_{s,PI}) - sin(L_{GSD-EF,PI})} \tag{14}$$

and scale the contributions of the respective NPP by the current area of the individual vegetation zone. The index $PI$ stands for
15    reference PI conditions and $f_{CO2}$ for the CO$_2$ fertilisation factor. In the original configuration, this factor was set to 0.65, which

was in good agreement with results by Friedlingstein et al. (2006). However, a revision of this value in a model intercomparison study yielded a lower value of 0.37 to be a more suitable value for the terrestrial biosphere (Zickfeld et al., 2013; Eby et al., 2013) and thus has also been used in the present study. Analogously, the land biosphere methane production (LBMP) (see Shaffer et al., 2008) is now calculated separately for the three vegetation zones as well.

Now, the four conservation equations per carbon isotope ([12,13,14]C) (see Shaffer et al., 2008) have to be calculated for each vegetation zone separately. The losses for reservoir size of litter and soil were dependent on the mean global atmosphere temperature in Shaffer et al. (2008) for the uniform vegetation. In order to achieve a more realistic dependency of this process in the three vegetation zone scheme, we now approximate a mean atmosphere temperature for each vegetation zone separately by making use of the DCESS model latitudinal temperature profile expressed as a second order Legendre polynomial in sine

of latitude (Shaffer et al., 2008). This yields,

$$T_{TF} = \frac{(T_{atm,LL} - 0.5 \cdot T_{atm,HL}) \cdot sin(L_{TF-GSD}) + 0.5 \cdot T_{atm,HL} \cdot sin(L_{TF-GSD})^3}{sin(L_{TF-GSD})}, \tag{15}$$

$$T_{GSD} = \frac{(T_{atm,LL} - 0.5 \cdot T_{atm,HL}) \cdot (sin(L_{GSD-EF}) - sin(L_{TF-GSD})}{sin(L_{GSD-EF}) - sin(L_{TF-GSD})} \tag{16}$$

$$+ \frac{0.5 \cdot T_{atm,HL} \cdot (sin(L_{GSD-EF})^3 - sin(L_{TF-GSD})^3)}{sin(L_{GSD-EF}) - sin(L_{TF-GSD})} \tag{17}$$

and

$$T_{EF} = \frac{(T_{atm,LL} - 0.5 \cdot T_{atm,HL}) \cdot (sin(L_{snow/ice}) - sin(L_{TF-GSD}))}{sin(L_{snow/ice}) - sin(L_{TF-GSD})} \tag{18}$$

$$+ \frac{0.5 \cdot T_{atm,HL} \cdot (sin(L_{snow/ice})^3 - sin(L_{TF-GSD})^3)}{sin(L_{snow/ice}) - sin(L_{TF-GSD})}. \tag{19}$$

Here, $T_{atm,LL}$ denotes the mean atmosphere temperature in the DCESS model low-mid latitude sector ($0° - 52°$) and $T_{atm,HL}$ in the model high latitude sector ($52° - 90°$). $L_{snow/ice}$ stands for the minimum of the latitude of the snow and the ice sheet line (see next section). Now, $\lambda_Q$, which influences the decay of litter and soil, can be calculated for each vegetation zone separately

with $\lambda_Q^i \equiv Q_{10}^{(T^i - T_{PI}^i)/10}$, where the index $i = 1, 2, 3$ stands for the three vegetation zones TF, GSD and EF. The conservation equations for the land biosphere reservoirs of $^{12}$C from Shaffer et al. (2008) for leaves ($M_G$), wood ($M_W$), litter ($M_D$) and soil ($M_S$) thus split into twelve equations, four for each vegetation zone:

$$\frac{dM_G^i}{dt} = \frac{35}{60} \cdot NPP^i - \frac{35}{60} \cdot NPP_{PI}^i \cdot \frac{M_G^i}{M_{G,PI}^i} \tag{20}$$

$$\frac{dM_W^i}{dt} = \frac{25}{60} \cdot NPP^i - \frac{25}{60} \cdot NPP_{PI}^i \cdot \frac{M_W^i}{M_{W,PI}^i} \tag{21}$$

$$\frac{dM_D^i}{dt} = \frac{35}{60} \cdot NPP^i \frac{M_G^i}{M_{G,PI}^i} + \frac{20}{60} \cdot NPP_{PI}^i \cdot \frac{M_W^i}{M_{W,PI}^i} - \frac{55}{60} \cdot NPP_{PI}^i \cdot \lambda_Q^i \cdot \frac{M_D^i}{M_{D,PI}^i} \tag{22}$$

$$\frac{dM_S^i}{dt} = \frac{5}{60} \cdot NPP^i \frac{M_W^i}{M_{W,PI}^i} + \frac{10}{60} \cdot NPP_{PI}^i \cdot \lambda_Q^i \cdot \frac{M_D^i}{M_{D,PI}^i} - \frac{15}{60} \cdot NPP_{PI}^i \cdot \lambda_Q^i \cdot \frac{M_S^i}{M_{S,PI}^i} \qquad (23)$$

Analogously, these equations are extended for the rare carbon isotopes $^{13}$C and $^{14}$C, where fractionation factors for land photosynthesis and, for $^{14}$C, radioactive sinks are considered (Shaffer et al., 2008). The flux of carbon dioxide between the terrestrial biosphere and the atmosphere is then determined by

$$F_{CO_2} = \sum_{i=1}^{3} -NPP^i + \frac{45}{55} \cdot NPP_{PI}^i \cdot \lambda_Q^i \frac{M_D^i}{M_{D,PI}^i} + \frac{15}{60} \cdot NPP_{PI}^i \cdot \lambda_Q^i \frac{M_S^i}{M_{S,PI}^i} \qquad (24)$$

As indicated above, $M_D^i$ and $M_S^i$ represent the biomass carbon reservoirs in litter and soil for the different vegetation zones, respectively, and $dM_D^i/dt$ and $dM_S^i/dt$ their decay rates. For the two rare carbon isotopes, additionally the corresponding fractionation factors $^{13,14}\alpha$ have to be considered. The flux is then given by

$$F_{^{13,14}CO_2} = \sum_{i=1}^{3} -NPP^i \cdot \frac{^{13,14}C}{^{12}C} \cdot {}^{13,14}\alpha + \frac{45}{55} \cdot NPP_{PI}^i \cdot \lambda_Q^i \cdot \frac{^{13,14}M_D^i}{^{13,14}M_{D,PI}^i} + \frac{15}{55} \cdot NPP_{PI}^i \cdot \lambda_Q^i \cdot \frac{^{13,14}M_S^i}{^{13,14}M_{S,PI}^i}. \qquad (25)$$

## 2.4 Formulation of permafrost

On glacial-interglacial time scales, global temperature changes lead to terrestrial ice sheet advances and retreats. These can cover large parts of the terrestrial biosphere and thereby prevent land-atmosphere carbon exchange in these areas. In the DCESS model, we account for this by introducing the parameter $L_{ice}$, that limits the poleward extent of the EF vegetation zone. During interglacials, when ice sheets retreat poleward to about 70° latitude, the poleward boundary of this zone is taken to be the equatorward extent of permafrost. For simplicity, we assume this extent to be the latitude of our model equatorward snow cover extent, $L_{snow}$, defined by the latitude at which global mean atmospheric temperature is 0°C in our zonally-averaged model. Hence, the minimum of these two parameters ($L_{snow,ice} = min(L_{snow}, L_{ice})$) at the current time step is used to determine the limitation of the EF vegetation zone. When $L_{snow/ice}$ advances and retreats on large spatial scales, organic carbon is buried/released below/from permafrost areas or areas below terrestrial ice sheets. That means that, additional land-atmosphere carbon ($^{12,13,14}$C) flux variations due to the changes of permafrost/ice sheet area are considered. For this, we add the permafrost flux term $^{12,13,14}F_{CO_2,PF}$ to Eqs. 24 and 25, which is calculated by

$$^{12,13,14}F_{CO_2,PF} = \frac{dA_{snow/ice}}{dt} \cdot {}^{12,13,14}C_{PF}. \qquad (26)$$

$$\frac{dA_{snow/ice}}{dt} = 2\pi R \cdot \left[ \left( 1 - \frac{270}{360} \right) \cdot \left( (1 - sin(L_{snow/ice}^t)) - (1 - sin(L_{snow/ice}^{t-1})) \right) \right] \qquad (27)$$

and denotes the change in snow or ice covered area. For this, $L_{snow/ice}$ of the previous ($t-1$) and the current ($t$) time step is taken. R denotes the Earth radius and the factor $(1 - 270/360)$ takes account for the land fraction in the model geometry.

$^{12}C_{PF}$, the amount of carbon being stored in permafrost, was approximated to $30\,kg \cdot m^{-2}$ by Schuur et al. (2015). Mainly due to the spatial heterogeneity of permafrost area and organic carbon content in permafrost soils, for example some peatland areas contain more than $100\,kg \cdot m^{-2}$, others far less than $30\,kg \cdot m^{-2}$, this value bears large uncertainties (see e.g. Zimov et al., 2009; Crichton et al., 2014). In a sensitivity experiment, we therefore also apply a doubled permafrost carbon content.

As shown in Zimov et al. (2009), carbon release rates from permafrost for warming are rapid with time scales on the order of 100 years. Such time scales are comparable to those of extra-terrestrial forest reoccupation of areas freed from permafrost, a process that we also take to be "instantaneous" in the model. On the other hand, carbon buildup in permafrost during cooling is a much slower process (Zimov et al., 2009). However, the model application in the present study starts from LGM conditions, following 80,000 years of cooling. Thus, we feel that this very simplified permafrost approach should be able to capture the first order effects of permafrost on carbon cycling during deglaciation. In fact, when we reduce atmospheric temperatures, the new vegetation scheme reacts with a vegetation decrease (in opposite to the old scheme) and thereby a $pCO_2$ increase, which again increases temperatures. Despite its simplicity, the permafrost implementation therefore helps to generate glacial conditions through its land carbon storage.

For the stable $^{13}C_{PF}$ isotope, carbon is buried and released through permafrost with the same isotope ratio. In our simulations, a typical mean isotope ratio for EF soil is $\delta^{13}C = -24‰$ (Zech (2012) estimates this value to $-27‰$). Using Eq. S9 of the Supplement, this yields a value of $0.33\,kg \cdot m^{-2}$ for permafrost $^{13}C$ given the above described assumption of permafrost $^{12}C=30\,kg \cdot m^{-2}$. For the doubled permafrost carbon experiment, this simply results in a doubled $^{13}C$ permafrost content. For $^{14}C_{PF}$, however, radioactive decay $(T_{1/2}(^{14}C) \approx 5730\,a)$ across glacial periods, when large parts of the high latitudes are covered by terrestrial ice sheets, has to be considered. While being buried with the current isotope ratio of soil, we therefore assume carbon to be released from permafrost radiocarbon free ($\Delta^{14}C = -1000‰$). This has also been considered to be reasonable by Zech (2012) for the last deglaciation. Land area uniformly covers 25% of the globe from the equator to 70 degrees latitude in the one hemisphere, DCESS model. For our model last glacial termination, permafrost affects latitudes between $47°$ and around $54°$ (see Fig. S3 in the Supplement and Sect. 3.1 for explanations), and is estimated as a two hemisphere mean. Across these latitudes, the land fraction averaged over both hemispheres is around 30% (see e.g. Matney, 2012). Thus, we did not deem necessary to further scale the permafrost effect due to global mean land fraction.

## 2.5 Evaluation of the new module

As a test of to what extent the newly developed land biosphere scheme adequately represents the behaviour of the land biosphere for global climate changes, we now present some detailed evaluation of the new module. With the old, simplified vegetation scheme, the DCESS model responds to cooling with an increase in land biomass. The terrestrial remineralisation rate decreases with sinking temperatures and hence, more carbon can be stored below ground. However, LGM reconstructions show less carbon in the land biosphere than for warmer, pre-industrial conditions (Peng et al., 1998; Prentice et al., 2011). This simplistic model behaviour can be seen in Fig. 3a, which shows the steady state terrestrial biomass as a function of $pCO_2$ and $T_{glob}$. These results are generated through prescribing various $pCO_2$ and $T_{glob}$ values in numerous 2 ka model simulations. In the new version (Fig. 3b), biomass decreases when temperatures sink as vegetation types shift and the snow line moves equatorward

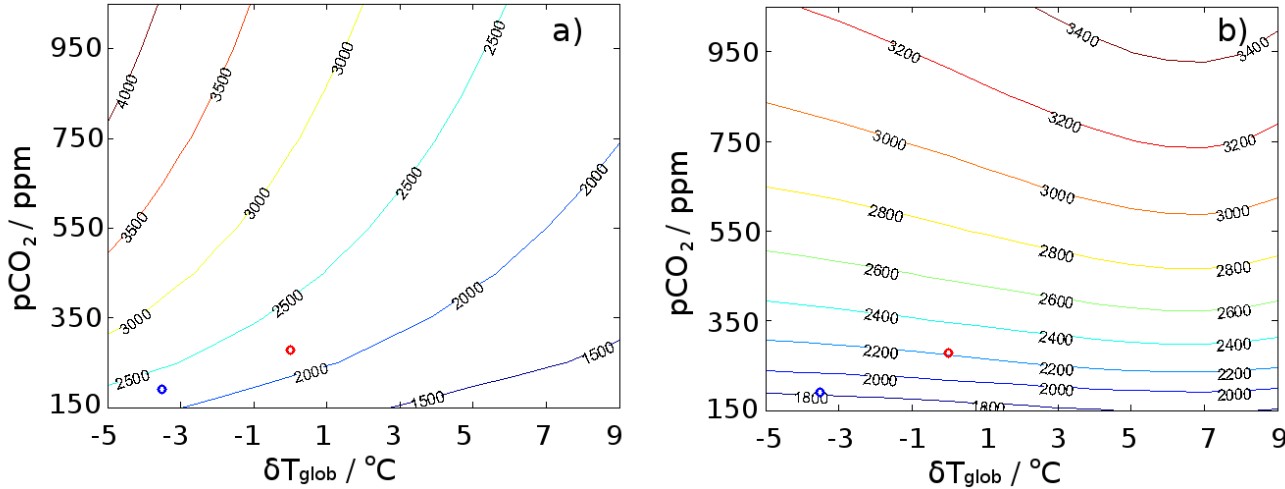

**Figure 3.** Steady state land biomass (GtC) as a function of global mean temperature (°C) and $pCO_2$ (ppm) deviations from the calibrated PI value for a) the old uniform biosphere scheme and b) the new biosphere scheme with three vegetation zones. The red circles denote PI and the blue circles LGM conditions.

(note however that a prescribed ice sheet line is not included in these simulations). The permafrost biomass, however, increases in the course of that process (not shown). The results in Fig. 3 show that the general land carbon storage is represented more realistically in the new model version.

Futhermore, we show the response of the model vegetation zones and the different vegetation reservoirs to a reduction of atmospheric temperatures and pCO$_2$ to LGM conditions and compare the results with complex vegetation models as well as with data reconstructions. To evaluate the vegetation scheme for LGM conditions, we carried out cooling simulations with the new and with the old biosphere scheme. For these, we started from a PI steady-state ($T_{glob} = 15°C$, $pCO_2$=280 ppm), but prescribed the global mean temperature to $T_{glob}$=11.5 °C (see Shakun et al., 2012) and the atmospheric pCO$_2$ concentration to 190 ppm (e.g. Monnin et al., 2001). A third cooling simulation was conducted with the new biosphere scheme and conditions as described but with an additional prescription of the ice sheet line to 47° latitude (see Sect. 3.1). Fig. 4a shows the global sum of total land biomass (LB) carbon (without carbon stored in permafrost) for the three cooling simulations as well as LB carbon for the three individual vegetation zones and Fig. 4b shows LB carbon of the vegetation reservoirs above ground (leaves + wood) and below ground (litter + soil) for the three simulations. We integrated these simulations over 2 ka to reach equilibrium

As already presented in Fig. 3, the cooling experiment again demonstrates that LB carbon increases in the old model version and decreases with the new biosphere scheme. Fig. 4b shows that the unrealistic increase of LB carbon is due to an increase in litter/soil carbon (i.e. biosphere below ground). In the simulation with the new biosphere scheme, this does not happen. The EF zone is dominated by biosphere below ground and due to the limitation of the poleward expansion of the EF zone through the snow line, this carbon reservoir is now decreasing. Also, the figures show that the timing of the change is represented more

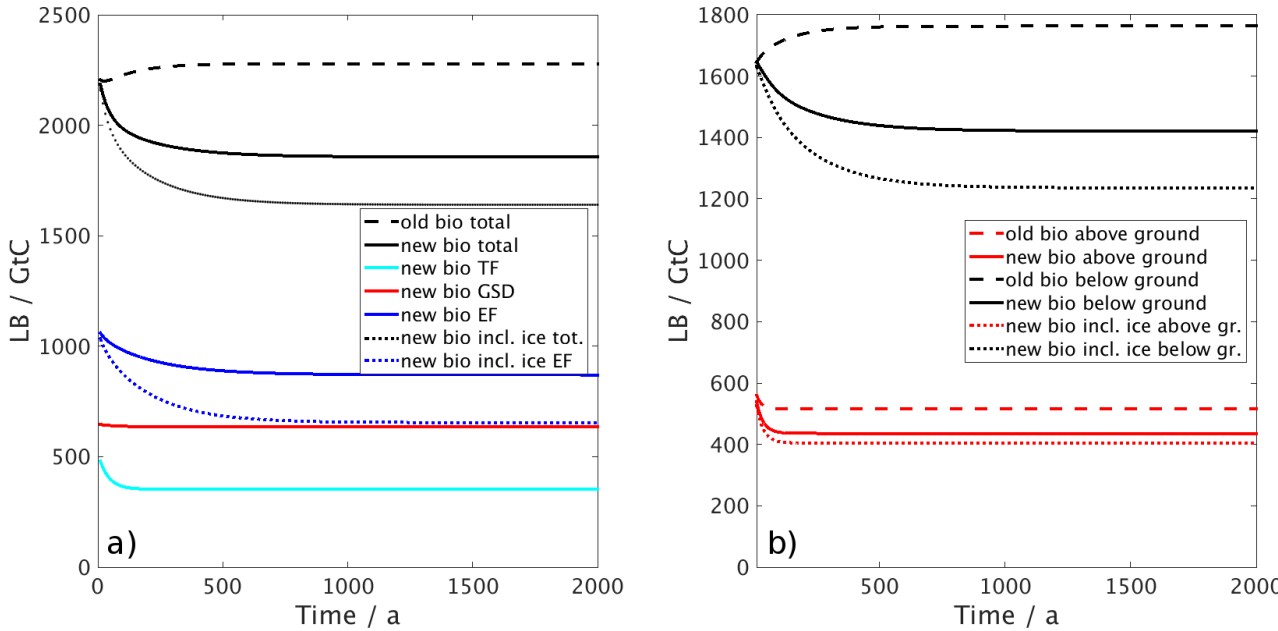

**Figure 4.** Cooling simulation (see text) for the model version with the old (dashed lines) and the new vegetation scheme without (solid lines) and with prescribed ice sheet expansion (dotted lines). a) Total land biomass carbon (in black) and separated into the three vegetation zones (TF: cyan, GSD: red, EF: blue) for the new vegetation scheme. b) Land biomass carbon separated into the reservoirs above ground (leaves + wood) in red and below ground (litter + soil) in black.

nuanced with the new biosphere scheme. The biospheres in the three vegetation zones show different reaction times according to their distinct temperatures and the dominating pool of vegetation in the respective area. When we also include the expansion of ice sheets ($L_{ice} = 47°$, for explanation see Sec. 3.1), covering larger areas of the EF vegetation zone than the snow line, the total land carbon pool decrease is stronger (dotted lines). It is mainly the biosphere below ground, exclusively in the EF zone,

5   that accounts for this.

A poleward limitation of the biosphere in the old vegetation scheme also leads to a reduction of LB carbon in the cooling simulation. To test this, we performed an additional simulation with the old vegetation scheme, but with the crude vegetation area limiting approach $A = (sin(L_{snow})/sin(L_{snow,PI}))^2$. In this cooling experiment the total LB carbon decreases, but not as much as with the new biosphere scheme and the decrease happens faster than with the new biosphere scheme (not shown).

10   The LB change in the EF zone mainly depends on variations in soil, which has a slow response time and is the largest biomass reservoir (Fig. 4a). The TF zone adapts much quicker to the new climate conditions because in this vegetation type the biomass is dominated by leaves and wood. This shows that not only the quantitative, but also the temporal description of land biomass changes is represented more accurately now. The GSD vegetation zone shows the smallest change in biomass, because in the cooling simulations the area of this vegetation zone changes only slightly, but rather just shifts latitudinally.

We calibrated the latitudinal dependency of the vegetation zone borders to match the LPJ model results. However, the calculation of carbon stored in the terrestrial biosphere at different climate conditions also depends on other parameters. Hence, we also evaluate the performance of the new DCESS vegetation scheme by comparing it to the results of the LPJ model study by Gerber et al. (2004). For this, table 2 shows the percentual change of biomass carbon in the cooling experiment for the new
vegetation scheme with and without ice sheet prescription and for the old vegetation scheme with and without the biosphere area limit (old bio plus) as well as for the LPJ model.

| $\Delta$LB / % | Total | Litter + Soil | Leaves + Wood |
|---|---|---|---|
| Old bio | +3.0 | +9.5 | −14.5 |
| Old bio plus | −10.8 | −5.2 | −26.0 |
| New bio | −18.0 | −14.2 | −28.5 |
| New bio ice | −27.6 | −25.3 | −33.6 |
| LPJ | −24.8 | −24.7 | −25.0 |

**Table 2.** Percentual change of biomass carbon in the cooling experiment for total biomass and divided into reservoirs above and below ground. DCESS model with the old biosphere scheme, with and without the crude approach for vegetation area limiting (see text), and with the new biosphere scheme with and without prescribed ice sheet expansion, and LPJ model study presented in Gerber et al. (2004).

This comparison demonstrates that in relation to the LPJ model, the adaptation of the LB to different climate conditions is captured much better with the new biosphere scheme. While with the old model version, biomass carbon increased, the new biosphere scheme produces most of the change that the LPJ model shows. Most of the improvement in LB variations through
the new vegetation scheme is due to the snow line, that limits the poleward expansion of the biosphere. Using the old biosphere with additional vegetation area limitation, LB carbon decreases under LGM climate conditions. However, with the new vegetation scheme, the snow line particularly limits the EF zone and this largely improves the overall representation of biomass below ground. When vegetation area reduction is applied to the old biosphere module, the biomass change above ground was already in good agreement with the LPJ model. Hence, the reason for the much larger changes in overall biomass between
the old and the new model version as shown in Fig. 4 is mainly due to the better representation of the slow change of the soil biomass in the EF zone. This more accurate representation of soil in the EF zone, however, is also due to the fact that now the biomass reservoir of each vegetation zone depends on the specific temperature of the zone in question and not on the global mean temperature as in the old model version. The prescribed ice sheet line at 47° latitude generates a further drawdown of the land carbon stock. The percentual change is then close to the LPJ model, about 3% higher. Vegetation above ground changes
too much, although this type of vegetation is not affected much by the ice line (see Fig. 4b), but due to the low total amount, the percentual change is high.

Peng et al. (1998) provide an overview of various studies that estimate the reduction in global land biomass for the LGM compared to present day. Those are based on either global climate model simulations, marine carbon isotope data changes or vegetation mapping approaches. Altogether, these studies show a large spread from 0 (Prentice and Fung, 1990) to $-1350\,GtC$ (Adams et al., 1990). The majority of the studies show values between $-300$ and $-700\,GtC$, a more recent modelling study by Prentice et al. (2011) provides values of $-550$ to $-694\,GtC$. Through the implementation of the new vegetation scheme, the DCESS model biomass carbon change between PI and LGM does improve from $+43$ to $-408\,GtC$. Indeed when this calculation includes the effect of ice sheet extent to $47°$, the LGM biomass change is $-626\,GtC$, in excellent agreement with the estimates cited above. Carbon stored in permafrost is around $600\,Gt$ for PI conditions and around $1000\,Gt$ for LGM climate when the ice sheets are included. Hence, the total amount of carbon on land is about $2800\,Gt$ for PI and $2600\,Gt$ for the LGM.

Ciais et al. (2012) estimate the LGM global carbon stock to $3640\pm400\,Gt$, so somewhat higher than in our model. Parts of this is due to our estimation of $30\,GtC \cdot m^{-2}$ for permafrost from Schuur et al. (2015). This apparent underestimation of the permfrost carbon inventory has to be kept in mind when analysing the model results and it will be addressed in the following with a sensitivity experiment using $60\,GtC \cdot m^{-2}$ for permafrost, for which the total amount of LGM carbon on land will be about 3600 Gt.

Overall, it can be stated that the new biosphere scheme with the three vegetation zones constitutes a significant improvement for the representation of the terrestrial biomass as well as the estimates of the size and timing of carbon exchanges between the terrestrial biosphere and the atmosphere. This new implementation better captures the complex interactions between the terrestrial and the atmospheric carbon exchange as is required for a better understanding of the processes that determine climate changes on glacial-interglacial time scales.

## 3   Application to Last Glacial Maximum and deglaciation

As a first application of the new DCESS terrestrial biosphere module, we simulate the deglaciation after the LGM, when global atmospheric temperatures rose by around 3.5 K (Shakun et al., 2012) and atmospheric $p$CO$_2$ increased from 190 ppm during the LGM to Holocene conditions of 260 ppm in a series of steps (e.g. Monnin et al., 2001). The most marked of these steps is a steep 38 ppm rise near the onset of the deglaciation, the Mystery Interval (Broecker and Barker, 2007). In the Supplement, we provide a literature review with details about the Mystery Interval including current hypotheses for the explanation of that climate change. Earlier studies found considerably greater LGM global mean cooling (Schneider von Deimling et al., 2006); recent estimates based on much improved temperature data, however, have shown LGM cooling of $3.2-4$ K (Schmittner et al., 2011; Shakun et al., 2012; Annan and Hargreaves, 2013).

A complete explanation for the $p$CO$_2$ and temperature increase at the onset of the last glacial termination must be able to reproduce a simultaneous decrease by 0.3‰ and 160‰ of atmospheric $\delta^{13}$C (Schmitt et al., 2012) and $\Delta^{14}$C (Reimer et al., 2013), respectively. Furthermore, it should also include how LGM deep water with high salinity (Adkins et al., 2002), low $\delta^{13}$C (Curry and Oppo, 2005) and $\Delta^{14}$C (Burke and Robinson, 2012) and low dissolved oxygen concentrations (but not widespread anoxia) (Jaccard et al., 2014) was formed during the last glacial. Hence, it requires the consideration of a globally compre-

hensive picture of the physical and biogeochemical processes in atmosphere, ocean and on land, as well as their interactions on various time scales. With its new biosphere scheme, the DCESS model is now better suited for investigations of that kind. However, a number of further adaptions need to be made to simulate LGM conditions and the transition to the Holocene. These are presented next followed by transient simulations across the last 25 kaBP. For these, the model was initialised and forced with the conditions described in Sect. 3.1. Since we focus on the MI ($17.5 - 14.5$ kaBP), we mainly present and discuss the time period from 20 to 10 kaBP. We assess the impact of various processes on the overall climate change with a focus on the new biosphere scheme and permafrost. In the process, we also evaluate proposed time series for the production of $^{14}$C in the atmosphere.

## 3.1    Model Last Glacial Maximum and transition

Guided by proxy-data records, we first modified several biogeochemical and physical parameters to generate a model steady-state that represents the LGM well. For this, a number of parameters can be considered as possible candidates (see e.g. Kohfeld and Ridgwell, 2009). However, under consideration of the possibilities provided by the enhanced model and knowledge about candidate parameters, we decided upon the adaptions described below.

Increased iron supply and thereby ocean fertilisation (Martin et al., 1990) through enhanced atmospheric dust concentrations during the LGM (see e.g., Mahowald et al., 1999, 2006b; Maher et al., 2010), particularly in the high southern latitudes (e.g. Lambert et al., 2013, 2015), probably led to enhanced new production of organic matter in the Southern Ocean (SO) by way of iron fertilization (see also Lamy et al., 2014; Martínez-García et al., 2014). To account for this, we modified the efficiency factor for new production in the model high latitude ocean sector from 0.36 (standard value for PI conditions, see Shaffer et al., 2008) to 0.5. This leads to a reasonable productivity increase of around 40% for the area of the SO and induces an atmospheric $p$CO$_2$ reduction of around 20 ppm, consistent with the DCESS model iron fertilisation results in Lambert et al. (2015). Moreover, an additional radiative effect of $-1\,Wm^{-2}$ (Mahowald et al., 2006a) for glacial conditions through atmospheric dust during the LGM is considered. For the transient simulations from the LGM to the Holocene, we have developed a transfer function between temperature and dust fluxes from proxy data records that we applied to the efficiency factor and to the radiative effect. It yielded an exponential dependency of dust with temperature; details can be found in the Supplement.

The lower sea level during the LGM (around 130 m, see e.g., Waelbroeck et al., 2002; Lambeck et al., 2014) and a thereby reduced ocean volume by around 3.5% (see e.g. Adkins and Schrag, 2002) is accounted for by increasing phosphate concentrations (the nutrient limiting source in the DCESS ocean biochemistry) and the ocean salinity (see Adkins et al., 2002) by 3.5%. For the transition of these parameters across the last 25 kaBP, we use the latest sea level reconstruction time series from Lambeck et al. (2014). We do not account for the expansion of land mass and vegetation due to reduction of sea level, which causes additional carbon storage (Joos et al., 2004). Although Joos et al. (2004) found that this effect is less important than the effect through climate/CO$_2$ caused vegetation changes or the ice sheet area effect, it can still have a considerable impact in deglaciation simulations and should be kept in mind when evaluating results. To generate LGM conditions for $\Delta^{14}$C in atmosphere and ocean, we applied the average cosmogenic $^{14}$C production rate from 25 to 26 kaBP ($PR_{14C} = 2.1 \cdot 10^4\, \text{atoms} \cdot \text{cm}^{-2}\text{s}^{-1}$).

For this and in most of the transient simulations, we use the most recent production rate time series developed by Hain et al. (2014). In a sensitivity analysis, the [14]C production rates from the studies by Laj et al. (2004) and Muscheler et al. (2004) are applied as well. A description of the main characteristics of these data is given in the Supplement.

LGM climate reconstructions show that the Laurentide ice sheet expanded as far south as 38°N (see e.g. Peltier, 2004). To account for this and the lack of an ice sheet in large parts of Siberia, and within the constraints of our zonally-averaged one hemisphere model, we prescribe the southernmost ice sheet extent to be 47°. For the transient simulations we impose the temporal retreat of the ice line to the disappearance of the ice sheets at 70° latitude during the Holocene. For this, we linearly prescribe $L_{ice}$ (see Sects. 2.3 and 2.4) to a data set presented in Shakun et al. (2012) showing the Northern Hemisphere ice sheet expansion from 100% (ice line at 47°) at the LGM to 0% (ice line at 70°) at present day. An example case for $L_{ice}$ and $L_{snow}$ in a transient simulation is given in the Supplement.

A model analogy to isolated deep water in the SO (see e.g. Watson and Naveira Garabato, 2006) is generated through application of a depth-dependent function for vertical exchange intensity in the high latitude ocean sector. For this, we impose a sharp decrease in vertical diffusion at around 1800 m ocean depth which limits mixing of the upper ocean layers with intermediate and deep ocean waters. The transition depth of this profile was varied to obtain LGM climate conditions that constrain all required oceanic and atmospheric variables. Through the application of this diffusivity profile, the isolated ocean waters below the transition change towards high dissolved inorganic carbon and alkalinity values as well as towards low oxygen concentrations and [13,14]C isotope ratios. This variation in vertical exchange intensity should not be understood as a change in real oceanic vertical diffusion, but rather as a model analogy for LGM conditions of the SO that were likely due to some combination of weakened or equatorward shifted westerly winds (Toggweiler and Russel, 2008; Anderson et al., 2009; d'Orgeville et al., 2010) and increased stratification through brine-induced effects (Bouttes et al., 2010, 2011; Mariotti et al., 2013). With its wide latitudinal extent and the land bounding poleward of 70°, the high latitude ocean sector of the DCESS model bears considerable resemblance to the SO. During the transient simulations, we slowly restore this modification back toward PI conditions between 17.5 and 14.5 kaBP to apply the entire effect of this process to the MI. In this process, deeper layers in the high latitude ocean sector are again brought in contact with surface layers promoting outgassing and ocean profiles go back toward the initial PI state shown in Shaffer et al. (2008). An illustration of the profile as well as a detailed technical description of the procedure and some additional information are presented in the Supplement.

When all these adaptations, plus a few minor changes (described in the Supplement), are applied, an 80 ka DCESS simulation leads to a steady climate state with conditions close to data-based LGM reconstructions. Atmospheric $pCO_2$ decreases to 187.9 ppm and the global mean atmosphere temperature to 11.70 °C. For $pCO_2$, proxy data records by Lüthi et al. (2008) provide a range of $186 - 198$ ppm and Shakun et al. (2012) present LGM global mean atmosphere temperatures between 11.5 and 11.8 °C. Moreover, atmospheric isotope ratios of $\delta^{13}C = -6.41‰$ and $\Delta^{14}C = 414.5‰$ and low oxygen values but no widespread anoxia in the deep ocean are achieved. This agrees well with proxy data records presented by Schmitt et al. (2012), Reimer et al. (2013) and Jaccard et al. (2014). An overview of these data and the ocean profiles for LGM conditions of various variables for the high and the low-mid latitude sector is given in the Supplement. In the following sections, we present analyses of the transient simulations from the LGM to the Holocene, using the transition functions described above.

## 3.2 Transient simulation results

To evaluate the impact of the individual new model developments, we carried out six transient simulations starting from LGM conditions as described above, varying the following parts of the model land biosphere: The first simulation features a nul-vegetation model (Nul_veg), meaning that the vegetation does not change from LGM conditions and land-atmosphere carbon (including the rare isotopes) fluxes are suppressed. Then, we use the old uniform land biosphere scheme (Old_bio) (including the snow/ice line-based reduction of biosphere area, see Sect. 2.5, but no permafrost parameterisation), and subsequently the new scheme without the permafrost (permafrost-atmosphere carbon fluxes set to to zero) and the new albedo features (NoPF_alb) and then the same but including the new albedo (NoPF). Last, we performed simulations with all the new model developments (REF) plus a further sensitivity experiment with a doubled ($60\,GtC{\cdot}m^{-2}$) permafrost carbon reservoir, as already mentioned above. An overview of these simulations is provided in table 3.

| Simulation | Long name | Setup |
|---|---|---|
| Nul_veg | Nul vegetation | Suppressed land-atmosphere fluxes |
| Old_bio | Old biosphere | Original uniform DCESS land biosphere scheme (No permafrost and no land area change) |
| NoPF_alb | No permafrost No albedo | Supressed fluxes from permafrost and old (not vegetation-dependent) albedo |
| NoPF | No permafrost | Supressed fluxes from permafrost |
| REF | Reference | Including all new developments as described in the text |
| 2xPF | Doubled permafrost | As REF but with two times the estimate for permafrost carbon reservoir |

**Table 3.** Overview of the DCESS model simulations with short description.

The results of these model simulations as well as data-based reconstructions are presented in Fig. 5 from 20 to 10 kaBP. As our transition functions (in particular the upwelling of the deep ocean) are tailor-made for simulating the Mystery Interval between 17.5 and 14.5 kaBP, we particularly focus in our analysis on this 3 ka slice of the last glacial termination.

The nul vegetation model shows the larges atmospheric changes across the MI. Uptake of carbon through the land biosphere does not take place in this simulation, therefore, all outgassed carbon stays in the atmosphere and amplifies global warming. This also reflects in the $\delta^{13}$C and $\Delta^{14}$C curves, isotopically strongly depleted carbon from the deep ocean decreases the atmospheric isotope ratios. Especially the far too strong drop in $\delta^{13}$C in the nul vegetation simulation points out that the regrowth of the biosphere and its preferential uptake of $^{12}$C keeps $\delta^{13}$C at a reasonable level in the other simulations, although the increase after 12 kaBP is not represented well in the model. The simulation with the old land biosphere scheme shows comparable changes across the MI, the missing expansion of the biosphere leads to a strong increase of atmospheric $p$CO$_2$. The change is even somewhat larger than in the nul-vegetation simulation, because the land biomass decreases due to the warming

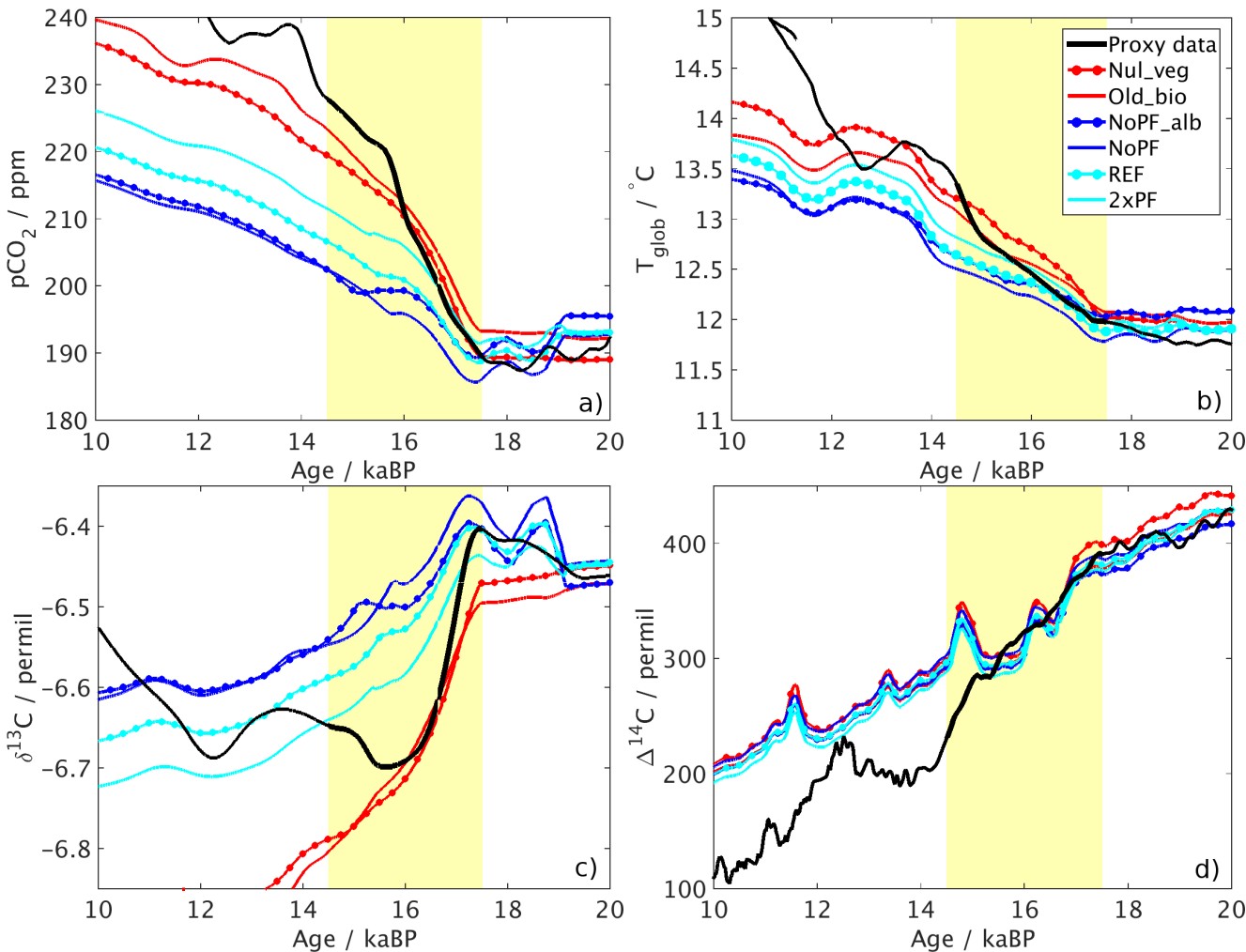

**Figure 5.** Atmospheric values for the DCESS simulations with nul vegetation model (Nul_veg., red line with dots), old biosphere scheme (Old_bio, red line), deactivated permafrost component and old albedo scheme (NoPF_alb, blue line with dots), deactivated permafrost component and new albedo scheme (NoPF, blue line), reference simulation with all new components (REF, light blue line with dots) and sensitivity experiment with doubled permafrost carbon reservoir (2xPF, light blue line), and data-based reconstructions (black); $pCO_2$ by Lüthi et al. (2008), temperatures by Shakun et al. (2012), $\delta^{13}C$ by Schmitt et al. (2012) and $\Delta^{14}C$ by Reimer et al. (2013).

(see also Sect. 2.5). Due to its reaction on vegetation changes, the new albedo diversification leads to stronger warming. This also generates some stronger $pCO_2$ increase. When we enable the permafrost parameterisation in the reference simulation, $pCO_2$ rises around 2.6 ppm more and the global mean atmosphere temperature around 0.1 °C. The results of the simulations start diverging at around 19 kaBP. This is when the change in ice sheet extent leads to first clear variations through its effect

5    on the permafrost parameterisation in the model (see Fig. S3 in the Supplement). The isotope ratios are only slightly affected

by these new features, in particular $\Delta^{14}C$ is controled mostly by the changes of the stratospheric production rate of $^{14}$C. The sensitivity experiment with a doubled permafrost reservoir shows a further increase of $p$CO$_2$. The difference between the 2xPF and the reference simulations is larger than between the reference and the NoPF simulation. The biosphere regrowth and its carbon uptake is only slightly enhanced in the simulation with doubled permafrost carbon reservoir. However, some more

change already happens before, i.e. after 19 kaBP. Therefore, this shows that uncertainties of that kind can have a considerable impact on climate change simulations. In comparison to data-based reconstructions, the MI atmospheric changes are closest in the 2xPF simulation (disregarding the Nul_veg and the Old_bio simulation). More than half of the $p$CO$_2$ and the global mean temperature changes are represented and the drop in $\delta^{13}$C is almost reached. $\Delta^{14}$C shows only little sensitivity to our new model developments.

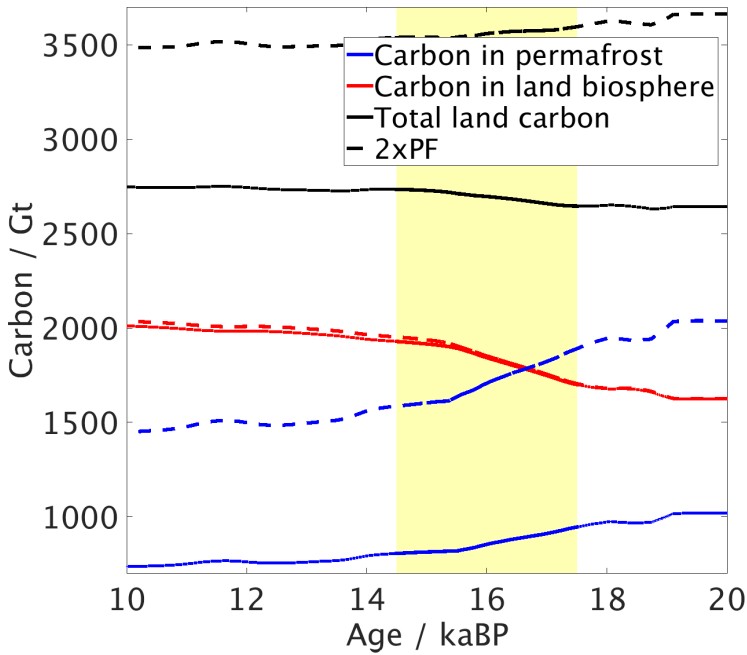

**Figure 6.** Carbon stored in soil below permafrost and in the terrestrial biosphere as well as their sum for the REF and for the 2xPF simulation.

Fig. 6 shows the changes of permafrost carbon, land biosphere carbon and their sum for the REF and the 2xPF simulations. In the reference simulation, carbon uptake through the regrowth of the biosphere across the MI slightly exceeds (by 70 GtC) carbon outgassing through ice sheet retreat and permafrost thawing then. In the 2xPF simulation, the permafrost carbon change slightly outweighs the vegetation effect. This demonstrates that the two mechanisms broadly compensate each other and provides an estimate of the uncertainty of the role of permafrost in our results. Also, model carbon release of 337 GtC

from permafrost is lower than that of Ciais et al. (2012), who found a 700 GtC difference between LGM and present day global permafrost carbon reservoir. Our lower estimate seems to be related to our simplified permafrost treatment and the simple assumption of 30 kg of available carbon per square meter of permafrost covered area (Schuur et al., 2008). The sensitivity

simulation with $60\,kgC \cdot m^{-2}$ in permafrost provides more realistic values for permafrost carbon release (667 GtC) and also for the global carbon reservoir ($\sim 3600$ GtC, see also Sect. 2.5).

In addition, we conducted four transient simulations to assess the impacts of the individual transition functions on atmospheric $T_{glob}$, $pCO_2$, $\delta^{13}C$ and $\Delta^{14}C$ changes (see Supplement). The transition functions described above were applied sequentially to better assess the impact of each process. These simulations show that during the 3 ka of the MI most of the simulated changes can be attributed to the resumption of the ocean high latitude vertical diffusion and the thereby induced outgassing of the carbon-rich and isotopically depleted deep waters. Our DCESS simulations reproduce only some aspects of the early last deglaciation, while others are underestimated because important processes are either missing or not adequately represented.

As has been mentioned above, the change in $\Delta^{14}C$ during the MI in the reference simulation is not as large as in the data-based reconstructions and not sensitive to our new developments of the land biosphere. Apart from atmospheric $CO_2$ itself and the release of deep ocean waters, $\Delta^{14}C$ is strongly influenced by the cosmogenic production rate of $^{14}C$. This production rate is determined with rather large uncertainties and there are different ways to derive it. In the Supplement, we show the three $^{14}C$ production rate time series of the studies by Laj et al. (2004); Muscheler et al. (2004) and Hain et al. (2014) across the last 25 kaBP. Here, we present an evaluation of the three $^{14}C$ production rate data applied to the reference simulation. In Fig. 7, we show the simulations with the three different production rates, as well as for a simulation with constant LGM-value production rate (Mus_PR, Muscheler et al. (2004) **p**roduction **r**ate; Laj_PR, Laj et al. (2004) **p**roduction **r**ate; LGM_PR, constant **LGM**-value **p**roduction **r**ate). The proxy data record by Reimer et al. (2013) is also included in the figure.

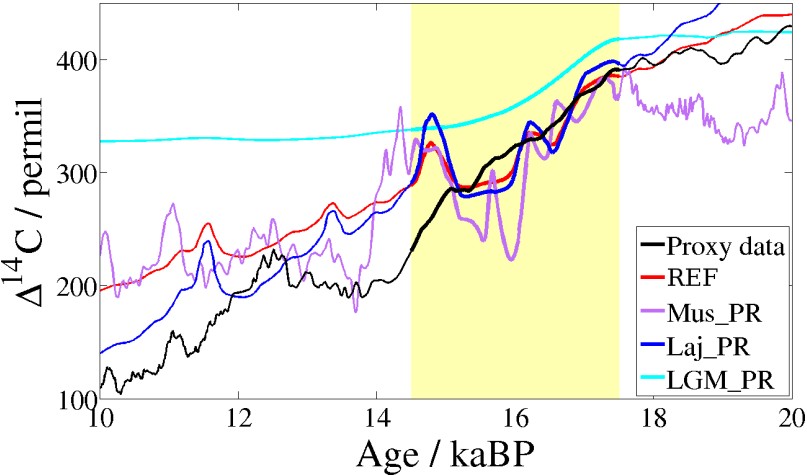

**Figure 7.** $\Delta^{14}C$ in transient simulations with all changes (see Sect. 3.1) applying different $^{14}C$ production rates from, Hain et al. (2014) (red, REF), Muscheler et al. (2004) (magenta, Mus_PR), Laj et al. (2004) (blue, Laj_PR) and fixed LGM production rate (cyan, LGM_PR), and data based reconstructions from Reimer et al. (2013) (black).

The simulation with constant $^{14}$C production rate at LGM level shows a $\Delta^{14}$C drop by $80\%o$ from the beginning to the end of the MI, almost entirely through the outgassing of isotopically depleted deep ocean waters. None of the $^{14}$C production rates can account for the remaining $80\%o$ reduction to explain the $\Delta^{14}$C decrease of $160\%o$ across the MI that can be seen in the data-based reconstruction by Reimer et al. (2013). With the data set by Hain et al. (2014), $\Delta^{14}$C drops by $96\%o$, using the Laj et al. (2004) data, a $105\%o$ decrease can be explained and the Muscheler et al. (2004) time series only leads to $-58\%o$ change. Furthermore, the proxy data does not show the production rate-caused variations within the MI and also, in the Mus_PR simulation, atmospheric $\Delta^{14}$C shows a large and sudden drop of around $150\%o$ shortly after the MI between 14.3 and 13.7 kaBP.

## 3.3   Discussion of transient simulations

The model reproduces more than half of the MI changes in atmospheric $pCO_2$, $T_{glob}$, $\delta^{13}$C and $\Delta^{14}$C as shown in data-based reconstructions. Overall, the representation of the land biosphere is shown to play an important role in the interplay of many processes. The model results reach from 12 to 31 ppm change in $pCO_2$ across the MI, i.e. from less than a third of the change presented in data-based reconstructions to more than 80%. The "best" results are reached for simulation using no biosphere fluxes or the original DCESS model biosphere module. However, these "best" results are obtained unambiguously for the wrong reasons. The missing uptake of carbon through the land biosphere for these simulations and the unrealistic behaviour of the old biosphere scheme to temperature changes lead to too high $pCO_2$ and temperature values. The $\delta^{13}$C isotope ratios reveal this model deficiency. $\delta^{13}$C further decreases in the nul-vegetation and old biosphere simulations after the MI, while in all other simulations, $\delta^{13}$C stagnates. In the data-based reconstructions, $\delta^{13}$C even rises again. Schmitt et al. (2012) mainly attribute this rise to the continuing regrowth of the land biosphere, which does not have such a strong effect on atmospheric $\delta^{13}$C in the model. According to Crichton et al. (2016), also peatlands could account for this effect, those however, are not included in our vegetation scheme. When we apply the double of the estimate of $30\,GtC \cdot m^{-2}$ from Ciais et al. (2012) the model results considerably improve in comparison with data-based reconstructions. In consideration of the apparent underestimation of total land biosphere carbon as shown in Sect. 2.5 and the large uncertainties in the estimation by Ciais et al. (2012), the usage of $60\,GtC \cdot m^{-2}$ is still reasonable.

The impact of the land biosphere on $\Delta^{14}$C is very small, even though we assume carbon released from permafrost to be radiocarbon free. The expected radiocarbon decrease generated through permafrost thawing can apparently be compensated by ocean-atmosphere exchange and subsequent mixing to the deeper ocean. It has to be considered that the carbon buried below permafrost seems to be underestimated in our model approach compared to a study by Ciais et al. (2012) and that interhemispheric see-saw effects can affect the timing of extensive permafrost ($^{14}$C depleted) carbon release, especially during Heinrich Event 1 (see e.g. Köhler et al., 2014). The much discussed sharp $\Delta^{14}$C drop of $160\%o$ (see Reimer et al., 2013) (note that in previous studies by Broecker and Barker (2007) or Reimer et al. (2009) this was referred to as $190\%o$) at the early stages of the last deglaciation is not entirely reproduced by this modelling study. By applying a constant LGM $^{14}$C production rate, all the above described processes can account for about $70\%o$ change. None of the three different time series of the $^{14}$C production rate can account for the rest of the $\Delta^{14}$C change. At most, the data by Laj et al. (2004) leads to an additional $25\%o$ decrease. However, the determination of the $^{14}$C production rate is obviously subject to large uncertainties. For example, the drop in the

Muscheler et al. (2004) time series at around 14 kaBP leads to a sudden $150\%_o$ decrease in $\Delta^{14}C$ in our model simulation but can not be seen in $\Delta^{14}C$ proxy data. In this context, it should be mentioned, that recent revisions to ice core time scales have not yet been applied for revising the reconstructed snow accumulation rates and $^{10}Be$ fluxes and its influence on the $^{10}Be$-based $^{14}C$ production rate (R. Muscheler, personal communication, 2015).

Most of the MI changes are caused by the upward transport of carbon-rich and isotopically depleted waters from the deep ocean through prescription of the vertical diffusion profile and its resumption. The dust component accounts for about 0.3 °C global temperature change during the MI. Since the other atmospheric quantities are only moderately affected by dust, most of that can be related to the direct dust radiative forcing. To account for the other half of changes that our simulations can not reproduce, several processes can be thought of being insufficiently represented in the model and moreover, this could also be

due to the timing of one or more of the transition functions, underrepresenting effects during the MI. Brovkin et al. (2007); Kohfeld and Ridgwell (2009) and Mariotti et al. (2013) discuss a number of processes that combined can account for the entire deglaciation, although with sometimes large uncertainties, not all of them were captured in our study. E.g., enhanced ocean remineralisation length scales during the glacial, due to less active bacteria at low temperatures, could trap more dissolved inorganic carbon in the deep ocean, which then could account for additional $CO_2$ outgassing but would also reduce deep ocean

dissolved oxygen concentrations. Also the volume of isolated deep waters in the SO is uncertain and moreover, water masses in other oceans may also have contributed to the overall atmospheric $pCO_2$ change (Rose et al., 2010; Okazaki et al., 2010; Kwon et al., 2012; Huiskamp and Meissner, 2012). The temperature and $pCO_2$ changes after the MI across the Bølling Allerød, the Younger Dryas and the Holocene are not expected to be simulated in detail by the DCESS model. Due to the model's simplified geometry, interactions between the hemispheres and thus the bipolar seesaw can not be represented. The simplicity of DCESS

model ocean dynamics also limit feedbacks of ocean-atmosphere interactions that may have contributed to the overall carbon cycle change during the MI. For instance, Mariotti et al. (2016) discuss the effect of North Atlantic freshening through ice sheet melting inducing upper water stratification and subsequent prevention of carbon uptake by the ocean that could contribute to enhanced $pCO_2$ at the end of the MI. An alternative approach would be to use 3-D modelling to deal specifically with one or more of the processes listed above. However, this would involve other types of uncertainties like the strength and position of

the Southern Westerly Winds and the parameterisation of diapycnal mixing.

## 4   Summary and conclusions

The land biosphere scheme that accounts for $^{12,13,14}C$ cycling with leaf, wood, litter and soil of the reduced complexity Earth System Model DCESS has been extended to three different vegetation zones. Based on a complex land biosphere model study, we defined dynamically varying vegetation borders on a global scale that depend on temperature variations. We also introduce

a parameterisation that accounts for carbon, including its rare isotopes, that is being trapped below the permafrost as well as below terrestrial ice sheets for glacial conditions and released during deglaciation events. In an evaluation, the new terrestrial biosphere scheme is shown to simulate more realistic global biomass size and timing in climate change experiments, and thereby significantly improves the representation of land-atmosphere carbon exchange rates in the DCESS model. For climate

change studies on glacial-interglacial time scales, these aspects can be crucial when analysing the contributions and interactions of processes controlling carbon exchange between land, atmosphere and ocean.

For a first application of the new biosphere parameterisation, the model is first tuned to Last Glacial Maximum conditions to subsequently carry out transient simulations across the last glacial termination. Along with a number of established adaptations of physical and biogeochemical parameters, the DCESS model successfully reproduces proxy data records of glacial conditions in the ocean and atmosphere when we impose the isolation of high latitude deep ocean waters. For the transient model simulations, we have additionally developed a set of explicit functions that describe the transitions of atmospheric dust, ocean volume and terrestrial ice sheet extent across the last 25 kaBP. These sensitivity eperiments show that large parts of the exceptional change in atmospheric $pCO_2$, $\delta^{13}C$, $\Delta^{14}C$ and $T_{glob}$ at the onset of the last glacial termination (Mystery Interval, 17.5-14.5 kaBP) can be represented by this approach. Some variations as seen in data-based reconstructions can not be reproduced by our model study. These remaining changes could possibly be captured by applying a dynamically more complex model including distinct water masses and a second hemisphere for representing bipolar seesaw effects, or by revising and/or adding one or more model parameterisations. New insights into these mechanisms can help to improve our understanding of global carbon cycle changes on centennial to millennial time scales.

The thawing of permafrost due to atmospheric warming and retreat of ice sheets, as well as the regrowth of the terrestrial biosphere, are found to play moderate, but important roles in explaining the climate change of this period of the last deglaciation. We found that these two processes broadly compensate each other in the model in terms of $CO_2$ exchange with the atmosphere, making little net contribution to atmospheric $pCO_2$ changes across the last transition. However, since our simulation bears considerable uncertainties, we also found that particularly the permafrost component could be underestimated. Simulations across the transition using the original DCESS land biosphere model also showed essentially no net contribution to atmospheric $pCO_2$ change as reflected in the very small change in land biomass between LGM and present day. But with the new biosphere module (including permafrost) this result is obtained in a more correct manner, in better agreement with proxy data and more complex modelling results.

Supplementary material related to this article is available online at doi:10.5194/gmd-0-1-2017-supplement.

*Code and data availability:* The DCESS model code is available at http://www.dcess.dk/ and all applied data are available as referenced.

*Acknowledgements.* We thank Raimund Muscheler for providing [14]C production rate data and information about it as well as Ricardo De Pol-Holz for discussions. This work was financed by Chilean Nucleus NC120066. GS and NA acknowledge support by FONDECYT grants # 1120040 and # 1150913, MR by Fondecyt grant # 1131055, and FL by Fondecyt grant # 1151427.

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
