# Peer review of "An improved land biosphere module for use in the DCESS Earth System Model (Version 1.1) with application to the last glacial termination"

_Geoscientific Model Development, 2016_

## Short Comment (SC1) · 20 Jan 2017

Dear authors,

in my role as Executive editor of GMD, I would like to bring to your attention our Editorial version 1.1:

http://www.geosci-model-dev.net/8/3487/2015/gmd-8-3487-2015.html

This highlights some requirements of papers published in GMD, which is also available on the GMD website in the 'Manuscript Types' section:

http://www.geoscientific-model-development.net/submission/manuscript_types.html

In particular, please note that for your paper, the following requirements have not been met in the Discussions paper:

- "The main paper must give the model name and version number (or other unique identifier) in the title."

In order to simplify reference to your developments, please add a model name (and/or its acronym) and a version number in the title of your article in your revised submission to GMD.

Yours,

Astrid Kerkweg

---

## Short Comment (SC2) · 24 Jan 2017

Comment on "An improved land biosphere module for use in reduced complexity Earth System Model with application to the last glacial termination"

By K.A.Crichton

The study presents developments of the DCESS earth system model, for vegetation zones and for a permafrost carbon pool. They present some validation for the vegetation zones, and then go on to perform and discuss the simulation of the last glacial termination. I focus on the permafrost module here.

The amount of carbon stored in the area defined as permafrost in the model is

30kg/m2, an approximation from present-day near surface soil organic carbon data in Schuur et al 2015. The approach to define where is the permafrost, is to use the latitude Lsnow (or Lice, whichever is lower) at the 0degC global temperature (page 9 line 6: is this a typo? Do you mean Lsnow is at the 0degC latitude? Perhaps this needs to be re-written).

Whilst this would indeed create a dynamic pool of carbon sensitive to changes in the area of Lsnow/ice, I am not convinced that this is a good representation of permafrost-carbon. If the concentration in this pool is fixed at 30kg/m2, then it cannot be properly taking account of the long time-to-equilibrium that would be seen in a permafrost-like carbon pool. Low accumulation and low decay rates means that the rate of change of area becomes important for soil carbon content. This is true for both release from "thawed" (i.e. no longer in Lice/snow) or newly permafrost (i.e. new Lice/snow) areas. They assume that this 30kg/m2 is instantaneous for new areas, and is instantly released in thawed areas (is this the case in the model?). As such it is entirely dependent on the parametrisation of Lsnow (and not soil carbon dynamics or decay rates).

The mean value of 30kg/m2 also does not take account of the true spatial heterogeneity of carbon content in permafrost soils. For example, some areas which underlay peatlands contain on the order of 100kg/m2+ organic carbon contents, and others far less than 30kg/m2. The spatial location of higher or lower-than-mean carbon content soils would make a big difference to carbon release rates. However, this is not possible to treat in this model (due to how it is set up) but should be mentioned.

I understand that this is a reduced complexity model, but it is important to incorporate accumulation and decay rates into a permafrost carbon pool model. This is especially true if the aim is to consider changing climates. At least this needs to be discussed in the text. See Zimov et al 2009 for example of the dynamic response of a permafrost-carbon model soil.

The authors say they tuned their model to a last glacial climate state, but this doesn't

appear to include the amount of carbon on land. Although the total change of -408GtC from LGM to PI is in alignment with recent estimates, the starting point at LGM from 1800GtC (fig 1) is far lower than Ciais et al 2012 estimate (at 3640+-400GtC). The authors do state p18 line10 that their permafrost pool is underestimated (compared to Ciais et al 2012), but they need to quantify this. It makes a big difference to the LGM simulation discussion. For example, with a far larger LGM permafrost carbon pool it is unlikely that regrowth of the terrestrial biosphere would compensate permafrost thawing for land carbon flux. It would also pull their LGM-PI land carbon change out. This needs to be discussed.

The authors have put a lot of effort into improving the land biosphere module for vegetation zones, but the permafrost pool representation is less well developed and not well explored, and is not validated. They need to consider whether representing permafrost carbon dynamics in this way, assuming an instant equilibrium with climate of soil carbon, is appropriate (I think it's not). And if not, then to instead develop a separate permafrost model for use with DCESS.

Zimov, N. S., et al. "Carbon storage in permafrost and soils of the mammoth tundra-steppe biome: Role in the global carbon budget." Geophysical Research Letters 36.2 (2009).

———————————————

---

## Referee Comment (RC1) · Anonymous Referee #1 · 13 Mar 2017

General comments

The paper presents a new terrestrial carbon cycle module within a the DCESS Earth System Model. Specifically, the authors expand the model, by accounting 3 vegetation zones, that can expand and contract, depending on global mean temperatures. The authors show, that the inclusion of these zones indeed creates a vastly different total vegetation carbon pool for glacial/interglacial transitions. They then couple the model with ocean and atmosphere content to evaluate the evolution of DELTA 14CO2 delta 13CO2, as well as CO2 concentration in the atmosphere during the last deglaciation.

I appreciate the work showing that indeed expanding and shrinking of areas of plant growth does have a significant effect and can indeed cause differences in the overall

response to global temperature change. However, the authors seem to be caught in a conundrum when applying their model to glacial-interglacial change. On the one hand, they try to discuss the degree with which the change in atmospheric carbon proxy can be reproduced by their model, but they have to deal with the result that these changes are mostly the imprint of ocean dynamics and ocean carbon cycle. As a result, there is much back and forth in the paper between discussing the terrestrial biosphere module and the entire model – leading to some confusion. Perhaps a way to remedy the whole thing is to organize the results from the DCESS without model improvement, talk about what how the transition is set up and carry out glacial-interglacial simulation in absence of a terrestrial module. Having this out of the way the focus can remain on the terrestrial system. Thus, first focus is glacial interglacial change with ocean/atmosphere boundary/initial conditions, and perhaps run a simulation without any vegetation change. In a next step one can then compare against this null model with the crude DCESS terrestrial module (no vegetation zones) against the improvement in the land model. The comparison may also not just discuss the outcome of carbon cycle and its impact on the prediction (and feedback) of temperature, but it could also include albedo effects due to the different biomes as well (and methane? - The authors mention also a wetland module at one point). The focus would then not be so much on the degree with which the DCESS model can reproduce CO2, but how the land parameterization affects DCESS dynamics.

It is important to note that the terrestrial biosphere likely gains carbon. Thus it works against the accumulation of CO2 in the atmosphere. In a way, your model improvement makes the situation worse. I think the author should point that out more clearly (despite the fact that the paleoclimate community is very well aware of). It also shows an important conundrum in modeling: Namely, even if you improve a model (and you know it), the outcome gets worse – which does not mean your model took a turn to the worse. In fact, such work help to foster continued model development.

Overall, I think the development and application of an improved terrestrial module in

a reduced complexity Earth System Model is a worthy endeavor. These types of model can offer great insight since they can easily be modified and interpretation is much more straightforward. I am sure the presentation can be modified that the paper achieves this goal by focusing on how the modification affect the overall Earth system dynamics.

Specific comments: Is methane considered as radiative forcing (methane emissions from terrestrial systems are briefly mentioned)? Also, is there a specific climate sensitivity applied to the model?

One of the limitations of the Gerber et al., 2004 study was, that they did not incorporate ice sheet, nor did they calculate potential effects of a reduced sea level. In particular, the reduction of sea level caused an additional and significant storage of carbon because of the expansion of the land mass. This may be worthwile discussing – see e.g. Joos et al., 2004 for applications with LPJ – against which the comparison here has been made.

It seems to me central parameters to glacial/interglacial change are lambda_Q (the Q10 factor), and fCO2 (the CO2 fertilization factor). Would it be worthwile to test the sensitivity of these in the DCESS outcomes? Method section: Parts of it seems to be result: I believe the simplistic model behavior should be juxtaposed with the improvement in the model section. In particular figure 1 should not appear in the method section, but be actually part of the results.

Equations 1 and 2: This is a 5th degree polynomial, is there a justification to use 5 degrees, can the extent not adequately represented by 3 degrees? I think it may be important to keep the number of parameters low in a reduced complexity Earth system model.

Equations 14-18: It seems these equations do not balance the carbon flux e.g. 10/60 of the leaf loss is unaccounted for. Also I don't understand equation 18: the units seem to be off and I don't see where the 45/55 comes from: In my understanding equation

18 should be the sum of equations 14-17.

I have trouble understanding how you calculated permafrost release. There are 2 numbers, one considers a permafrost storage of 30 kg m-2, but what is the 0.33 kg m-2? And how is this number linked to the isotope ratio? This may be my limitation, but perhaps there are ways to clarify this.

Figure 6: It is not clear what the production rate is in the ALL_TF simulation (red line).

Discussion of transient simulations: A great deal of this discussion focuses on ocean carbon cycling, which is not surprising given that the ocean dominants the carbon cycle on this time scale. However, there is little support to the items raised in the paper. Where is it detailed out, how much each of the radiative forcing contributes to the temperature increase (dust etc.), and how this affect isotopic distributions. In some instances, it may be sufficient to point to the appropriate figure/text in the supplementary material, but perhaps it is also worthwile considering additional plots. And again, I suggest some restructuring to better set apart the overall mechanisms of glacial/interglacial carbon cycling from the discussion of the improved vegetation dynamics.

P9L23. Starting from "As is,. . ." until the end of paragraph, this seems to be misplaced.

P10L4: Please also state what the initial global temperature is (14 degree C?)

P2L12: Check abbreviation for extratropical forest

Reference Joos, F., S. Gerber, I. C. Prentice, B. L. Otto-Bliesner, and P. J. Valdes. 2004. Transient simulations of Holocene atmospheric carbon dioxide and terrestrial carbon since the Last Glacial Maximum. Global Biogeochemical Cycles 18:GB2002.

––––––––––––––––––––––

---

## Referee Comment (RC2) · Anonymous Referee #2 · 21 Mar 2017

General comments

The authors present an improved land biosphere module which is then used in a reduced-complexity Earth system model to simulate the last glacial termination with a focus on carbon cycle changes. Although the processes causing the $CO_2$ rise during the last glacial termination are far from being understood and contributions to increase this understanding are highly welcome, I have some major concerns about this paper.

I'm not very convinced by the structure of the paper. The paper focuses on the description of an improved land biosphere component on the one side and on the coupled Earth system model response during the last glacial termination on the other side. Much of the changes in the global carbon cycle in the model during the last termination are due to changes in the ocean carbon cycle resulting mostly from prescribed transition functions. It is therefore not very clear what the message of the paper is supposed to be. Focusing only on the carbon cycle changes driven by the land and how the improvements to the land model affect the simulated land carbon response during deglaciation would probably result in a more straightforward message being delivered to the readers.

The authors claim that they have improved the land model but the improvements are discussed only in a very qualitative manner. Several quantities could be compared with observed or reconstructed (also model-based) values, e.g. permafrost area (both present day and LGM) and permafrost carbon content (present day), NPP (LGM).

A great advantage of a simple model over more complex models is the lower computational cost. This strength could be exploited to perform some parameter sensitivity analysis which would help to understand how robust the presented results are to changes in unconstrained parameter values. I'm a bit disappointed that this has not been done in the paper.

It is questionable if a progress in understanding the role of land carbon changes during glacial termination can be attained by using the extremely simplified model described in this study for several reasons:

1) In the model, permafrost carbon reacts instantaneously to changes in the snow-ice line. This seems a quite crude parameterization and neglects the long time scales associated with permafrost carbon dynamics. The assumption of a uniform permafrost carbon concentration of 30 kgC/m2 is not supported by observations which show large spatial variations in permafrost carbon over Siberia. At least a sensitivity analysis to this value would be appropriate.

2) The Northern Hemisphere ice sheet extent at LGM is strongly dependent on longitude, with the Laurentide ice sheet over North America extending as far south as 50° while Siberia was ice-free. The implications of this asymmetry, which can not be

considered in a zonally averaged model, should at least be discussed in the paper.

3) How the polynomial relations for the latitude of the borders between vegetation zones are derived from Fig. 4 in Gerber et al. (2004) is not very clear. Since this is supposed to be a technical paper, some more details could be given. On what quantity is the separation between vegetation zones based? What is the justification for using a 5-th order polynomial? Also, what is the zone north of the snowline considered to be?

For the LGM cooling relative to preindustrial the IPCC gives a very likely range of 4-7°C cooling, while a value 0f 3.5°C is used in the model based on Shakun et al. 2012. This is only one example where a sensitivity analysis would be appropriate. I would expect the choice of global temperature at LGM to have a large impact on the simulated land carbon storage at LGM.

Specific comments

The last part of the last sentence on page 2 would fit into the abstract.

I would suggest moving the discussion of Figure 1 (sentences on page 3, lines 25-27 and 31-34) to section 2.4.

In the caption of Table 1, 'globally averaged for one hemisphere' should be replaced with e.g. 'integrated over one hemisphere'. (And why not give the global values instead of hemispheric values? That would make the values more easily interpretable.)

Page 5, line 4: what do the authors mean by 'latitude of 0°C global mean temperature'?

In section 2.1, first the separation of vegetation zones should be described and only afterwards Table 1 should be discussed. The total area of each vegetation zone should also be given together with the values of biomass reservoirs and NPP in Table 1.

Page 9, line 4 and 7: 'BF' -> 'EF'

Page 12, line 3: 'agree well WITH other estimates'.

Page 12, lines 5: I can't see how the authors can say something about improvements in the 'timing' of carbon exchanges between land and atmosphere based on the results presented in the evaluation section.

————————————————

---

## Author Comment (AC1) · 17 Apr 2017

**Reply to:**
**SC1: 'Executive Editor Comment on "An improved land biosphere module for use in reduced complexity Earth System Models with application to the last glacial termination"', Astrid Kerkweg, 20 Jan 2017**

Dear authors, in my role as Executive editor of GMD, I would like to bring to your attention our Editorial version 1.1: http://www.geosci-model-dev.net/8/3487/2015/gmd-8-3487-2015.html This highlights some requirements of papers published in GMD, which is also available on the GMD website in the 'Manuscript Types' section: http://www.geoscientific-model-development.net/submission/manuscript_types.html

In particular, please note that for your paper, the following requirements have not been met in the Discussions paper:

- "The main paper must give the model name and version number (or other unique identifier) in the title."

In order to simplify reference to your developments, please add a model name (and/or its acronym) and a version number in the title of your article in your revised submission to GMD.
Yours,
Astrid Kerkweg

Dear Astrid Kerkweg,
thank you for your comment. To address this we have now modified our title to read: "An improved land biosphere module for use in the DCESS Earth System Model (Version 1.1) with application to the last glacial termination"

---

## Author Comment (AC2) · 17 Apr 2017

**Reply to:**
*thank you for your comment. Please find our answers (in blue) to your comments (in black) below:*

[Figure]

The study presents developments of the DCESS earth system model, for vegetation zones and for a permafrost carbon pool. They present some validation for the vegetation zones, and then go on to perform and discuss the simulation of the last glacial termination. I focus on the permafrost module here.

The amount of carbon stored in the area defined as permafrost in the model is 30kg/m2, an approximation from present-day near surface soil organic carbon data in Schuur et al 2015. The approach to define where is the permafrost, is to use the latitude Lsnow (or Lice, whichever is lower) at the 0degC global temperature (page 9 line 6: is this a typo? Do you mean Lsnow is at the 0degC latitude? Perhaps this needs to be re-written).

To clarify we modified this text to read "During interglacials when ice sheets retreat poleward, the poleward boundary of this zone is taken to be the equatorward extent of permafrost. For simplicity we assume that this extent is defined by $L_{snow}$ , the latitude of $0°C$ global mean temperature".

Whilst this would indeed create a dynamic pool of carbon sensitive to changes in the area of Lsnow/ice, I am not convinced that this is a good representation of permafrost-carbon. If the concentration in this pool is fixed at 30kg/m2, then it cannot be properly taking account of the long time-to-equilibrium that would be seen in a permafrost-like carbon pool. Low accumulation and low decay rates means that the rate of change of area becomes important for soil carbon content. This is true for both release from "thawed" (i.e. no longer in Lice/snow) or newly permafrost (i.e. new Lice/snow) areas. They assume that this 30kg/m2 is instantaneous for new areas, and is instantly released in thawed areas (is this the case in the model?). As such it is entirely dependent on the parametrisation of Lsnow (and not soil carbon dynamics or decay rates). The mean value of 30kg/m2 also does not take account of the true spatial heterogeneity of carbon content in permafrost soils. For example, some areas which underlay

peatlands contain on the order of 100kg/m2+ organic carbon contents, and others far less than 30kg/m2. The spatial location of higher or lower-than-mean carbon content soils would make a big difference to carbon release rates. However, this is not possible to treat in this model (due to how it is set up) but should be mentioned.

I understand that this is a reduced complexity model, but it is important to incorporate accumulation and decay rates into a permafrost carbon pool model. This is especially true if the aim is to consider changing climates. At least this needs to be discussed in the text. See Zimov et al 2009 for example of the dynamic response of a permafrost-carbon model soil.

In the simplified context of the DCESS model, mean values and zonal boundaries characterise each of our three new vegetation zones and for consistency this approach was extended to our treatment of permafrost. As shown in Zimov et al. 2009, carbon release rates from permafrost for warming are rapid with time scales on the order of 100 years. Such time scales are comparable to those of extra-terrestrial forest reoccupation of areas freed from permafrost, a process that we also take to be "instantaneous" in the model. Thus there is also internal consistency in this point. On the other hand, carbon buildup in permafrost during cooling is a much slower process (Zimov et al. 2009) but we start our simulations from LGM conditions following 80,000 years of cooling. Thus we feel that our permafrost approach, although very simplified, should be able to capture the first order effects of permafrost on carbon cycling during deglaciation, as reported in our paper (but see also below). We will bring a discussion of these points more up front in our revision.

The authors say they tuned their model to a last glacial climate state, but this doesn't appear to include the amount of carbon on land. Although the total change of -408GtC from LGM to PI is in alignment with recent estimates, the starting point at LGM from 1800GtC (fig 1) is far lower than Ciais et al 2012 estimate (at 3640+-400GtC). The

authors do state p18 line10 that their permafrost pool is underestimated (compared to Ciais et al 2012), but they need to quantify this. It makes a big difference to the LGM simulation discussion. For example, with a far larger LGM permafrost carbon pool it is unlikely that regrowth of the terrestrial biosphere would compensate permafrost thawing for land carbon flux. It would also pull their LGM-PI land carbon change out. This needs to be discussed.

Ms. Crichton is mistaken in stating that our LGM starting point for the amount of carbon on land is 1800 GtC. Rather this is our result for our new biosphere but does not include our estimate for permafrost. When that estimate is included (see Fig. 5) the total amount of carbon on land is about 2800 GtC, much larger but still somewhat low compared to the Ciais et al. 2012 estimate of 3640+-400GtC. To address this and our apparent underestimation of the permafrost pool we will carry out an additional simulation as a sensitivity study whereby we will use a doubled permafrost content, i.e. 60 kg of carbon per square meter. For this case our total amount of carbon on land will be about 3800 GtC, in good agreement with the Ciais et al. 2012 estimate. We will include the results of this new simulation in our Figs. 4 and 5.

The authors have put a lot of effort into improving the land biosphere module for vegetation zones, but the permafrost pool representation is less well developed and not well explored, and is not validated. They need to consider whether representing permafrost carbon dynamics in this way, assuming an instant equilibrium with climate of soil carbon, is appropriate (I think it's not). And if not, then to instead develop a separate permafrost model for use with DCESS.

As follows from what we wrote above, we will put more effort into explaining, exploring and validating our treatment of permafrost.

Zimov, N. S., et al. "Carbon storage in permafrost and soils of the mammoth tundra-steppe biome: Role in the global carbon budget." Geophysical Research Letters 36.2 (2009).
* * *

---

## Author Comment (AC3) · 17 Apr 2017

**Reply to:**
**Interactive comment on An improved land biosphere module for use in re-
duced complexity Earth System Models with application to the last glacial
termination by Roland Eichinger et al. from Anonymous Referee #1**

*Dear Anonymous Referee #1,*

*thank you very much for your comments and suggestions. Please find our answers (in
blue) to your comments (in black) below:*

[Figure]

General comments

The paper presents a new terrestrial carbon cycle module within a the DCESS Earth System Model. Specifically, the authors expand the model, by accounting 3 vegetation zones, that can expand and contract, depending on global mean temperatures. The authors show, that the inclusion of these zones indeed creates a vastly different total vegetation carbon pool for glacial/interglacial transitions. They then couple the model with ocean and atmosphere content to evaluate the evolution of DELTA 14CO2 delta 13CO2, as well as CO2 concentration in the atmosphere during the last deglaciation.

I appreciate the work showing that indeed expanding and shrinking of areas of plant growth does have a significant effect and can indeed cause differences in the overall response to global temperature change. However, the authors seem to be caught in a conundrum when applying their model to glacial-interglacial change. On the one hand, they try to discuss the degree with which the change in atmospheric carbon proxy can be reproduced by their model, but they have to deal with the result that these changes are mostly the imprint of ocean dynamics and ocean carbon cycle. As a result, there is much back and forth in the paper between discussing the terrestrial biosphere module and the entire model - leading to some confusion. Perhaps a way to remedy the whole thing is to organize the results from the DCESS without model improvement, talk about what how the transition is set up and carry out glacial-interglacial simulation in absence of a terrestrial module. Having this out of the way the focus can remain on the terrestrial system. Thus, first focus is glacial interglacial change with ocean/atmosphere boundary/initial conditions, and perhaps run a simulation without any vegetation change. In a next step one can then compare against this null model with the crude DCESS terrestrial module (no vegetation zones) against the improvement in the land model. The comparison may also not just discuss the outcome of carbon cycle and its impact on the prediction (and feedback) of

temperature, but it could also include albedo effects due to the different biomes as well (and methane? - The authors mention also a wetland module at one point). The focus would then not be so much on the degree with which the DCESS model can reproduce CO2, but how the land parameterization affects DCESS dynamics.

We appreciate these suggestions and are now restructuring the paper to accommodate them, thereby putting focus squarely on the new vegetation model. As part of this we now will compare the following six simulations of the last glacial termination in one or two of our figures: Simulation 1 (S1)↦a nul model with no vegetation change but all other prescribed forcings, S2↦S1 but using the old DCESS veg model, S3↦S1 with the new DCESS veg model but no veg albedo effects nor permafrost, S4↦S3 but including new veg albedo; S5↦S4 but including permafrost ($30kgC/m^2$) and S6↦S4 but including increased permafrost ($60kgC/m^2$; see also answer to K. Crichton's comment). This creates the desired shift of focus towards the effects of the land parameterisation on DCESS model dynamics. The discussion section will be restructured accordingly.

It is important to note that the terrestrial biosphere likely gains carbon. Thus it works against the accumulation of CO2 in the atmosphere. In a way, your model improvement makes the situation worse. I think the author should point that out more clearly (despite the fact that the paleoclimate community is very well aware of). It also shows an important conundrum in modeling: Namely, even if you improve a model (and you know it), the outcome gets worse - which does not mean your model took a turn to the worse. In fact, such work help to foster continued model development.

We will discuss these points in more detail in the paper. As the data show and our model simulates, the terrestrial biosphere by itself likely gains carbon across the last termination. However, as shown in our Fig. 5, when permafrost is included there is an overall carbon loss that will increase for our new simulation with enhanced permafrost.

Overall, I think the development and application of an improved terrestrial module in a reduced complexity Earth System Model is a worthy endeavor. These types of model can offer great insight since they can easily be modified and interpretation is much more straightforward. I am sure the presentation can be modified that the paper achieves this goal by focusing on how the modification affect the overall Earth system dynamics.

Specific comments:

Is methane considered as radiative forcing (methane emissions from terrestrial systems are briefly mentioned)? Also, is there a specific climate sensitivity applied to the model?

Yes, methane is considered as radiative forcing. As described in Shaffer et al. (2008), our old land biosphere model was tuned to emit the methane required to balance atmospheric oxidation while achieving observed pre-industrial atmospheric methane concentrations. We have adopted this approach for our new land biosphere model too but we found that this simple extension of our earlier approach led to values for LGM methane considerably higher than observed in ice cores, about 500 ppb compared to 350 ppb. So we decided to use prescribed methane (and nitrous oxide) concentrations from ice core observations for our radiative forcing calculations in our last termination simulations. However, inspired by the reviewer's comment we are revisiting our methane calculations with the idea to only allow methane production in the "wet" vegetation zones (tropical or tropical/boreal). Depending on our new results we may then use simulated methane (rather than prescribed) for our radiative forcing in the simulations and then, of course, include a new short section describing our new

methane approach.

In our simulations we use a climate sensitivity of 2.5°C for a $pCO_2$ doubling as was explained and motivated in section S6 of our Supplement.

One of the limitations of the Gerber et al., 2004 study was, that they did not incorporate ice sheet, nor did they calculate potential effects of a reduced sea level. In particular, the reduction of sea level caused an additional and significant storage of carbon because of the expansion of the land mass. This may be worthwile discussing - see e.g. Joos et al., 2004 for applications with LPJ - against which the comparison here has been made.

In our work we do in fact include the ice sheet area effect, albeit in a simplified way. On the other hand we did not include possible effects of sea level change and associated land exposure. In Joos et al 2004, this effect was found to be considerably less important than the ice sheet area effect or the climate/$CO_2$ change effect. However, we will discuss this point as suggested.

It seems to me central parameters to glacial/interglacial change are lambda_Q (the Q10 factor), and fCO2 (the CO2 fertilization factor). Would it be worthwile to test the sensitivity of these in the DCESS outcomes?

These are of course central parameters in any land biosphere model but here we prefer not to go into sensitivity studies based around variations of these parameters. We feel that that would carry us too far afield. Furthermore, the values for the parameters we now use have proven to give comparable land biosphere results in recent intercomparison studies of past and future warming and $pCO_2$ change (Eby et al 2013, Climate

of the Past; Zickfeld et al 2013, Journal of Climate). When we participated in these studies we found that the original DCESS model fertilisation factor (0.62; Shaffer et al, 2008) appeared to be too high. By reevaluation of this we arrived at a lower value (0.37) that we used in the 2013 studies and continue to use today.

Method section: Parts of it seems to be result: I believe the simplistic model behavior should be juxtaposed with the improvement in the model section. In particular figure 1 should not appear in the method section, but be actually part of the results.

We agree that the section of land biosphere albedo, including Fig. S1, should have been included in the main text and in the revision we will do so. As mentioned above, we will also do a sensitivity study of the effect of land biosphere albedo.

Equations 1 and 2: This is a 5th degree polynomial, is there a justification to use 5 degrees, can the extent not adequately represented by 3 degrees? I think it may be important to keep the number of parameters low in a reduced complexity Earth system model.

The answer to this requires a more in-depth description, both here and in our revision, of how we arrived at these curves. Our point of departure was the total tree cover frame of Fig. 4 of Gerber et al. 2004, Global Biogeochemical Cycles. On that frame we read off, at 2°C intervals from -10 to 10°C deviation from pre-industrial global mean temperature, the latitudes in the Northern Hemisphere of 50% tree cover both above and below the subtropical zone of lower tree cover. Each of these two sets of 11 points formed the basis for our curve fitting. We found that 5th order polynomials provided good fits to each of these sets whereas 3rd order polynomials did not, in particular for the tropical/grassland boundary. In our revised manuscript we will plot the individual

"data" points in Figure 2.

Equations 14-18: It seems these equations do not balance the carbon flux e.g. 10/60 of the leaf loss is unaccounted for. Also I don't understand equation 18: the units seem to be off and I don't see where the 45/55 comes from: In my understanding equation 18 should be the sum of equations 14-17.

There is a mistake in Equation 16. The fraction just to the right of the equal sign should be 35/60 (not 25/60), explaining the imbalance pointed out by the reviewer, This is an error only in the written equation of the manuscript; the model code uses the correct equation. As stated in Shaffer et al, 2008, "...NPP is distributed between leaves and wood in the fixed ratio 35:25, all leaf loss goes to litter, wood loss is divided between litter and soil in the fixed ratio 20:5, litter loss is divided between the atmosphere (as $CO_2$) and the soil in the fixed ratio 45:10. Soil loss is to the atmosphere as $CO_2$...". This helps to explain the other fractions in equations 16 and 17. Unfortunately as pointed out by the reviewer, Equation 18 is in error as written down in the manuscript (but not in error in the corresponding model code equation). In the manuscript Equation 18 should read

$$F_{CO2} = \sum_{i=1}^{3} -NPP^i + \frac{45}{60} NPP_{PI}^i \cdot \lambda_Q^i \frac{M_D^i}{M_{D,PI}^i} + \frac{15}{60} \cdot NPP_{PI}^i \cdot \lambda_Q^i \frac{M_S^i}{M_{S,PI}^i}$$

As pointed out by the reviewer, Equation 18 should be the sum of equations 14-17. This was so in the model code and now is also the case in the manuscript equations. Furthermore, Equation 19 is now also being corrected in the same manner in the manuscript.

I have trouble understanding how you calculated permafrost release. There are 2 numbers, one considers a permafrost storage of 30 kg m-2, but what is the 0.33 kg

m-2? And how is this number linked to the isotope ratio? This may be my limitation, but perhaps there are ways to clarify this.

The number $0.33\,kg{\cdot}m^{-2}$ (actually 0.329....) is the amount of $^{13}C$ needed to yield a $\delta^{13}C$ of -24 in permafrost soil, given the assumption of a constant $30\,kg{\cdot}m^{-2}$ carbon in permafrost. It is calculated using Eq. S11 of the Supplement. For clarity we will include this information in the revised test.

Figure 6: It is not clear what the production rate is in the ALL_TF simulation (red line).

The ALL_TF simulation uses the production rate from Hain et al. (2014), as stated in the figure caption.

Discussion of transient simulations: A great deal of this discussion focuses on ocean carbon cycling, which is not surprising given that the ocean dominants the carbon cycle on this time scale. However, there is little support to the items raised in the paper. Where is it detailed out, how much each of the radiative forcing contributes to the temperature increase (dust etc.), and how this affect isotopic distributions. In some instances, it may be sufficient to point to the appropriate figure/text in the supplementary material, but perhaps it is also worthwile considering additional plots. And again, I suggest some restructuring to better set apart the overall mechanisms of glacial/interglacial carbon cycling from the discussion of the improved vegetation dynamics.

Through the above described restructuring of the paper, the discussion of ocean carbon cycling is now put more into the background. Furthermore, we now clearly state where individual items raised in the discussion are supported, basically that is Figure

S7 and Table S3. However, to maintain the focus of the paper on land carbon cycling, we refrain from adding more plots on these topics to the manuscript.

P9L23. Starting from "As is, ... " until the end of paragraph, this seems to be misplaced.

We may not have made clear enough why we include this text here. The model land fraction in this latitude range of changing permafrost extent is important since we want a global permafrost estimate but work with a model that represents the Earth with one generic hemisphere. If the mean land fraction of North and South Hemisphere (in that latitude range) was very different from the model land fraction, some scaling for the permafrost effect would be required. We now underline the importance of this point by revising the paragraph to read:

Land area uniformly covers 25% of the globe from the equator to 70 degrees latitude in the one hemisphere, DCESS model. For our model last glacial termination, permafrost affects latitudes between $47°$ and around $54°$ (see Fig. S3 in the Supplement), and is estimated as a two hemisphere mean. Across these latitudes, the land fraction averaged over both hemispheres is around 30% (see e.g. Matney, 2012). Thus we did not feel it to be necessary to further scale the permafrost effect due to global mean land fraction.

P10L4: Please also state what the initial global temperature is (14 degree C?)

The initial global temperature is $15°C$ and is now included in the text.

P2L12: Check abbreviation for extratropical forest

Corrected to EF in the text. Thanks again to the reviewer for careful reading of our manuscript.

―――――――――――――――――――

---

## Author Comment (AC4) · 17 Apr 2017

**Reply to:**
*thank you for your comments and suggestions. Please find our answers (in blue) to your comments (in black) below:*

General comments:

The authors present an improved land biosphere module which is then used in a reduced-complexity Earth system model to simulate the last glacial termination with a focus on carbon cycle changes. Although the processes causing the CO2 rise during the last glacial termination are far from being understood and contributions to increase this understanding are highly welcome, I have some major concerns about this paper.

I'm not very convinced by the structure of the paper. The paper focuses on the description of an improved land biosphere component on the one side and on the coupled Earth system model response during the last glacial termination on the other side. Much of the changes in the global carbon cycle in the model during the last termination are due to changes in the ocean carbon cycle resulting mostly from prescribed transition functions. It is therefore not very clear what the message of the paper is supposed to be. Focusing only on the carbon cycle changes driven by the land and how the improvements to the land model affect the simulated land carbon response during deglaciation would probably result in a more straightforward message being delivered to the readers.

In the revised version, the paper is being restructured to put the focus squarely on carbon cycle changes driven by the land and how land model improvements affect carbon cycling during deglaciation. We now start the results and the discussion from an additional simulation with constant land biosphere, followed by the old and the new biosphere model transition simulations and the discussion on permafrost impact. Thereafter, the evaluation of the impacts of individual transition functions for the transient simulation starts. This creates a shift of focus towards the role of the land biosphere on

climate and carbon cycling and the discussion of ocean carbon cycling is now moved more into the background. Please also see also our answers to reviewer 1.

The authors claim that they have improved the land model but the improvements are discussed only in a very qualitative manner. Several quantities could be compared with observed or reconstructed (also model-based) values, e.g. permafrost area (both present day and LGM) and permafrost carbon content (present day), NPP (LGM).

We do in fact compare our calculations for land and permafrost carbon storage change between pre-industrial and LGM to other estimates (Sections 2.4 and 3.2). In the revised version we are extending these comparisons to present day and LGM carbon stocks as well. For example, we now state that our total LGM amount of carbon on land is about 2800 GtC (land biosphere plus permafrost), which is somewhat low compared to the Ciais et al. 2012 estimate of 3640±400GtC. We discuss how this can lead to an underestimation of the permafrost effect in the transition simulation and carry out an additional sensitivity simulation with a doubled concentration for permafrost carbon (also see our above response to K. Crichton on this).

A great advantage of a simple model over more complex models is the lower computational cost. This strength could be exploited to perform some parameter sensitivity analysis which would help to understand how robust the presented results are to changes in unconstrained parameter values. I'm a bit disappointed that this has not been done in the paper.

As discussed in our response to reviewer 1, we prefer to retain our values for central parameters like $Q_{10}$ and $f_{CO2}$ since the values we use have proven to give comparable land biosphere results to those of other models in recent intercomparison studies of

past and future warming and $pCO_2$ change. Also as discussed above we are now doing additional sensitivity studies, for example using a doubled concentration for permafrost carbon.

It is questionable if a progress in understanding the role of land carbon changes during glacial termination can be attained by using the extremely simplified model described in this study for several reasons:

1) In the model, permafrost carbon reacts instantaneously to changes in the snow-ice line. This seems a quite crude parameterization and neglects the long time scales associated with permafrost carbon dynamics. The assumption of a uniform permafrost carbon concentration of 30 kgC/m2 is not supported by observations which show large spatial variations in permafrost carbon over Siberia. At least a sensitivity analysis to this value would be appropriate.

For this see the detailed response to K. Crichton on DCESS model structure, very different time scales for uptake and release of permafrost carbon and consistency with the treatment of the land biosphere. Furthermore, as mentioned several times earlier, we will be carrying out an additional sensitivity simulation using a uniform permafrost carbon concentration of 60 kgC/m$^2$.

2) The Northern Hemisphere ice sheet extent at LGM is strongly dependent on longitude, with the Laurentide ice sheet over North America extending as far south as $50°$, while Siberia was ice-free. The implications of this asymmetry, which can not be considered in a zonally averaged model, should at least be discussed in the paper.

In fact the Laurentide Ice Sheet extended as far south as $38°$N during the LGM. To

account for this and the lack of an ice sheet in large parts of Siberia and within the constraints of our zonally-averaged model, we prescribed the southernmost ice sheet extent to be 47° and used ice sheet reconstructions to estimate a time series of ice sheet extent retreat (see Section 3.1). We feel that this is a meaningful approach and the best that can be done to be consistent with the degrees of complexity in the other model modules. We will supply some more discussion on this in the revision.

3) How the polynomial relations for the latitude of the borders between vegetation zones are derived from Fig. 4 in Gerber et al. (2004) is not very clear. Since this is supposed to be a technical paper, some more details could be given. On what quantity is the separation between vegetation zones based? What is the justification for using a 5-th order polynomial? Also, what is the zone north of the snowline considered to be?

The first two of these questions are dealt with in detail in a response to reviewer 1, including new additions to the revised manuscript. In our simplified approach the zone poleward of the snowline (0°C) is taken to be permafrost. In our revised version we will mention this up front in Section 2.1 to avoid confusion.

For the LGM cooling relative to preindustrial the IPCC gives a very likely range of 4-7°C cooling, while a value 0f 3.5°C is used in the model based on Shakun et al. 2012. This is only one example where a sensitivity analysis would be appropriate. I would expect the choice of global temperature at LGM to have a large impact on the simulated land carbon storage at LGM.

Recent estimates based on much improved temperature data have shown LGM cooling of 3.2 - 4°C (Shakun et al. 2012; Schmittner et al., 2011, Science; Annan and Hargreaves, 2013, Climate of the Past). Earlier studies based on much less data found

considerably greater LGM global mean cooling (e. g. Schneider von Deimling et al., 2006; Geophysical Research Letters). Thus we feel that the choice of a 3.5°C LGM cooling is well motivated. Furthermore, our simulations use a modest climate sensitivity of 2.5°C based in part on the well-constrained LGM cooling we use (see Section S6 of our Supplement). Otherwise, our Fig. 1b provides a sensitivity analysis for our new land biosphere model like that proposed by the reviewer. The figure shows only a reduction in land biosphere carbon storage of about 50 GtC for a 5°C LGM cooling compared to a 3.5°C cooling. Furthermore, as shown in our Fig. 2, the model snow-line does not extend equatorward of 47° for a 5°C cooling such that our prescribed ice sheet extent continues to form our equatorward permafrost extent. In the model this translates to no change in permafrost carbon storage for a 5°C LGM cooling compared to a 3.5°C cooling. Some of the above will be included in the revision.

Specific comments:

The last part of the last sentence on page 2 would fit into the abstract. I would suggest moving the discussion of Figure 1 (sentences on page 3, lines 25-27 and 31-34) to section 2.4.

These are good suggestions and are being adopted in the revision.

In the caption of Table 1, 'globally averaged for one hemisphere' should be replaced with e.g. 'integrated over one hemisphere'. (And why not give the global values instead of hemispheric values? That would make the values more easily interpretable.)

We also follow the reviewer's suggestion here and are providing global values in Table 1 in the revision. The table caption is being revised to read: Table 1. Pre-industrial

distribution of carbon storage among model land carbon pools as well as model net primary production for the three vegetation zones (see Chapin et al., 2011, and citations therein).

Page 5, line 4: what do the authors mean by 'latitude of 0°C global mean temperature'?

In our model, a meridional temperature distribution in the atmosphere is obtained by fitting a second-order Lagrange polynomial to the atmospheric temperature of the low-mid latitude and high latitude boxes (see Shaffer et al, 2008). For clarification, in our revision we are modifying this text to read "During interglacials when ice sheets retreat poleward, the poleward boundary of this zone is taken to be the equatorward extent of permafrost. For simplicity we take this extent to be the latitude of our model equatorward snow cover extent, Lsnow , defined by the latitude at which global mean atmospheric temperature is 0°C in our zonally-averaged model".

In section 2.1, first the separation of vegetation zones should be described and only afterwards Table 1 should be discussed. The total area of each vegetation zone should also be given together with the values of biomass reservoirs and NPP in Table 1.

We are also adopting these reviewer suggestions in our revision.

Page 9, line 4 and 7: 'BF' -> 'EF'

Thanks, done.

Page 12, line 3: 'agree well WITH other estimates'.

This has now been corrected.

Page 12, lines 5: I can't see how the authors can say something about improvements in the 'timing' of carbon exchanges between land and atmosphere based on the results presented in the evaluation section

With the new vegetation scheme, a better quantitative description of the respective land biosphere pools is possible, including their individual reaction to temperature now calculated separately for each zone. As shown in Figure 3 for example, the TF zone vegetation reacts rapidly to temperature change since it is dominated by above-ground vegetation. Such differentiation was not possible with the old scheme and hence the timing of changes could not be represented as accurately as with the new scheme. We will try to make this clearer in the revised manuscript text.

---

## Author Response (AR2)

**AUTHOR'S RESPONSE**

Concerning the second revision of

*"An improved land biosphere module for use in the DCESS Earth System Model (Version 1.1) with application to the last glacial termination" by Eichinger, R., Shaffer, G., Albarrán, N., Rojas, M., and Lambert, F., 2017, Geoscientific Model Development (gmd-2016-306).*

Dear Dr. David Lawrence,

in consequence of the second revision of our manuscript, I have changed

- "latitude at which global mean atmospheric temperature is 0°C", to
  "latitude at which the zonal mean atmospheric temperature is 0°C."

- and in the Supplement on page S4 line 2
  "latitude of $T_{glob} = 0°C$", to
  "latitude of zonal mean atmospheric T= 0°C"

Below is also the marked up version (see page 9).

Best regards,

Roland Eichinger,

August 15, 2017

[revised manuscript text omitted]